# COMPOSING NOVEL CLASSES: A CONCEPT-DRIVEN APPROACH TO GENERALIZED CATEGORY DISCOVERY

## ABSTRACT

We tackle the generalized category discovery (GCD) problem, which aims to discover novel classes in unlabeled datasets by leveraging the knowledge of known classes. Previous works utilize the known class knowledge through shared representation spaces. Despite their progress, our analysis experiments show that impressive novel class clustering results are achieved in the feature space of a known class pre-trained model, suggesting that existing methods may not fully utilize known class knowledge. To address it, we introduce a novel concept learning framework for GCD, named ConceptGCD, that categorizes concepts into two types: derivable and underivable from known class concepts, and adopts a stage-wise learning strategy to learn them separately. Specifically, our framework first extracts known class concepts by a known class pre-trained model and then produces derivable concepts from them by a generator layer with a covariance-augmented loss. Subsequently, we expand the generator layer to learn underivable concepts in a balanced manner ensured by a concept score normalization strategy and integrate a contrastive loss to preserve previously learned concepts. Extensive experiments on various benchmark datasets demonstrate the superiority of our approach over the previous state-of-the-art methods. Code will be available soon.

## 1 INTRODUCTION

Despite the notable achievements of recent deep learning models (He et al., 2016; Dosovitskiy et al., 2020), most of them still face challenges in open-world scenarios when encountering novel concepts. In contrast, humans are able to leverage their existing knowledge to discover new concepts. Taking inspiration from this ability, Han et al. (2019; 2021) introduce the problem of Novel Class Discovery (NCD), which is further extended by Vaze et al. (2022a) to a more practical setting named Generalized Category Discovery (GCD), where unlabeled data include both known and novel classes. The unique interplay between labeled and unlabeled data in this problem setting presents a distinctive challenge: how can we effectively utilize labeled data to assist the model in learning novel classes?

In most GCD works (Vaze et al., 2022a; Wen et al., 2022; Zhang et al., 2022; Sun & Li, 2022; Wang et al., 2024a; Choi et al., 2024), the training paradigm involves amalgamating all data, whether labeled or unlabeled, into a unified learning process with a shared encoder to discover novel classes. However, as demonstrated in crNCD (Gu et al., 2023), the strategy of sharing encoders undermines meaningful class relations, complicating the transfer of knowledge between known and novel classes.

To further explore the impact of this shared strategy and better understand knowledge transfer in GCD, we embrace a prevalent hypothesis (Zeiler & Fergus, 2014; Allen-Zhu & Li, 2020a;b) that each class consists of certain concepts learned by neural networks and the responses of those concepts are used for class prediction. In the context of GCD, owing to the semantic relationship between known and novel classes, we assume that some novel class concepts can be derived from known class concepts via simple transformations, while others cannot (i.e., underivable). Intuitively, these derivable concepts are one of the key reasons why known class data can help the model learn novel class data in GCD problems. To study the impact of the derivable concepts on knowledge transfer, we construct a baseline as shown in Fig. 1, where a set of derivable concepts are first generated through a linear transformation applied to the known class concepts and then are used for classification and clustering in GCD. Interestingly, we observe that, as shown in Tab. 1, such designed concepts—a subset of all derivable concepts—yield competitive performance compared to state-of-the-art methods. To investigate this, we analyze encoder neuron activation patterns on 100 randomly selected samples,

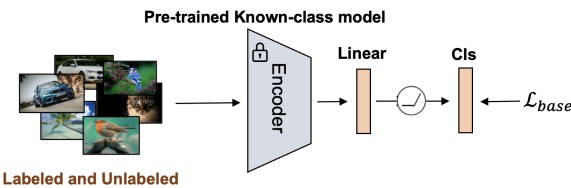

Figure 1: Generate derivable concepts. The linear layer and classifier are trained on novel and known class data with $\mathcal{L}_{base}$ (Wen et al., 2022) defined in Eq. 1.

Table 1: "crNCD" (Gu et al., 2023), "SPTNet" (Wang et al., 2024a) and "Linear" performance on novel class.

| Method | SPTNet | crNCD | Linear |
|---|---|---|---|
| CUB | **65.1** | 58.6 | 64.5 |
| Scars | 49.3 | 44.3 | **52.8** |
| Aircraft | **58.1** | 51.3 | 55.6 |
| ImgNet100 | 81.4 | 76.9 | **81.5** |
| Cifar100 | **75.6** | 70.6 | 69.5 |

Table 2: The statistical data of the minimal KL divergence between neuron responses in our linear method and those of crNCD and SPTNet. The interval represents the KL Divergence range.

| Method | (0, 0.01) | [0.01, 0.1) | [0.1, 0.2) | [0.2, 0.5) | [0.5, 1.0) | [1.0, ∞) |
|---|---|---|---|---|---|---|
| SPTNet | 13 | 264 | 181 | 190 | 77 | 43 |
| crNCD | 7 | 113 | 141 | 289 | 192 | 26 |

transforming them into probability distributions using softmax and computing the KL divergence between models (details in Appendix O). As shown in Tab. 2, our linear model exhibits distinct neuron activation patterns (KL divergence > 0.5) compared to SPTNet and crNCD. This finding suggests that SPTNet and crNCD fail to fully capture derivable concepts, which are crucial for model performance, as shown in Tab. 1. A potential cause is that these methods utilize a shared encoder to learn those concepts, and hence the known class knowledge in this encoder may be influenced by the noise introduced by novel class label uncertainty, leading to low-quality derivable concepts.

Based on these insights, we propose a novel concept learning approach, named ConceptGCD, that partitions class concepts into two categories—derivable and underivable from known class concepts—and learns them in a stage-wise manner. To this end, we introduce an expandable encoder that first focuses on learning known class concepts and then generates derivable concepts based on these known class concepts, followed by a third stage that learns underivable concepts. This stage-wise learning strategy ensures that the learning of derivable concepts can be isolated from the noisy learning of underivable concepts, thus effectively leveraging known class knowledge to discover novel classes.

Specifically, our novel ConceptGCD comprises three core steps: 1) **Learn known class concepts**. We train a deep network model on the labeled known class data as our pre-trained known-class model to obtain known class concepts. To ensure that the model captures a broad range of concepts, we introduce a concept covariance loss, which encourages independence between the different concepts. 2) **Generate derivable concepts**. We employ a linear layer and a ReLU layer after the encoder of the pre-trained known-class model as our derivable concept generator, which is trained on known and novel class data with a covariance-augmented loss. 3) **Learn underivable concepts**. The final stage focuses on learning underivable concepts while preserving the previously generated concepts. To do so, we first expand the dimension of the original linear layer to capture new concepts and then learn those concepts with a contrastive loss in the feature space. Moreover, we introduce a concept score normalization to balance the model responses across new and previous concepts, which prevents the model from over-relying on the derivable concepts and reduces the impact of noisy learning.

To validate our approach, we conduct extensive experiments across six standard benchmarks. Our method demonstrates substantial improvements over the current state of the art, thereby highlighting the efficacy of our framework. Furthermore, our experimental analysis offers clear evidence of the contribution of each component in our method. Our contributions can be summarized as follows:

- We introduce a novel concepts learning framework, named ConceptGCD, for GCD that efficiently generates concepts from known classes using a simple generator layer, and learns independent concepts through the expanded generator layer in a subsequent stage.

- We are the first to incorporate a covariance loss in GCD, which promotes diversity among the learned concepts. Additionally, we propose a novel concept score normalization technique to ensure a more balanced learning of different concepts.

- We conduct extensive experiments on several benchmarks to validate the effectiveness of our method, which outperforms the state-of-the-art by a significant margin.

## 2 RELATED WORK

The problem of NCD is formalized in Han et al. (2019), aiming to cluster novel classes by transferring knowledge from labeled known classes. Specifically, KCL (Hsu et al., 2018a) and MCL (Hsu et al., 2018b) use the labeled data to learn a network that can predict the pairwise similarity between two samples and use the network to cluster the unlabeled data. Instead of using pairwise similarity to cluster, DTC (Han et al., 2019) utilizes the deep embedding clustering method (Xie et al., 2016) to cluster the novel class data. Later works mostly focus on improving the pairwise similarity (Han et al., 2021; Zhao & Han, 2021), feature representations (Zhong et al., 2021a;b; Wang et al., 2024b; Liu et al., 2024), or clustering methods (Fini et al., 2021; Zhang et al., 2023; Xu et al., 2024).

Recently, Vaze et al. (2022a) extended Novel Class Discovery into a more realistic scenario where the unlabeled data come from both novel and known classes, known as Generalized Category Discovery (GCD) (Rastegar et al., 2023; Wang et al., 2024a). To tackle this problem, GCD (Vaze et al., 2022a) adopts semi-supervised contrastive learning on the pre-trained visual transformer (Dosovitskiy et al., 2020). Meanwhile, ORCA (Cao et al., 2022) proposes an uncertainty adaptive margin mechanism to reduce the bias caused by the different learning speeds on labeled data and unlabeled data. Later, most works (Sun & Li, 2022; Zhang et al., 2022; Pu et al., 2023; Vaze et al., 2024) focus on designing a better contrastive learning strategy to cluster novel classes. For example, PromptCAL (Zhang et al., 2022) uses auxiliary visual prompts in a two-stage contrastive affinity learning way to discover more reliable positive pairwise samples and perform more reasonable contrastive learning. DCCL (Pu et al., 2023) proposes a dynamic conceptional contrastive learning framework to alternately explore latent conceptional relationships between known classes and novel classes, and perform conceptional contrastive learning. However, those methods typically rely on transferring knowledge implicitly by sharing encoders, which can be restrictive as shown in Gu et al. (2023). Specifically, Gu et al. (2023) distill knowledge in the model's output space which contains limited information compared to representation space, and their method cannot be applied to the GCD setting directly due to its special design of weight function. In contrast, we introduce an innovative concept learning framework that generates and learns novel concepts, thereby explicitly extracting and transferring knowledge from known classes to assist novel class discovery within the rich representation space.

## 3 METHOD

### 3.1 PRELIMINARIES

**Problem formulation.** In the Generalized Category Discovery (GCD) problem, the dataset is composed of a labeled known classes set $\mathcal{D}^l = \{x_i^l, y_i^l\}_{i=0}^{|\mathcal{D}^l|}$ and an unlabeled set $\mathcal{D}^u = \{x_j^u\}_{j=0}^{|\mathcal{D}^u|}$, which contains both known and novel classes. Here $x, y$ represents the input image data and the corresponding label. In addition, we denote the number of known and novel classes as $N^k$ and $N^n$, and assume $N^n$ is known (Vaze et al., 2022a; Zhang et al., 2022). The goal is to classify known classes and cluster novel classes in $\mathcal{D}^u$ by leveraging $\mathcal{D}^l$.

**Basic Loss.** Among almost all existing GCD methods (Vaze et al., 2022a; Wen et al., 2022; Zhang et al., 2022; Sun & Li, 2022; Choi et al., 2024), the basis of these models can be succinctly deconstructed into two components. The first component is the supervised learning of the labeled known class data. The second component is the unsupervised learning of both known and novel class unlabeled data. Therefore, the core part of their final loss function can be written as:

$$\mathcal{L}_{base} = (1 - \alpha)\mathcal{L}_s + \alpha\mathcal{L}_u, \tag{1}$$

where $\mathcal{L}_s$ is the supervised learning loss on labeled data, and $\mathcal{L}_u$ is the unsupervised learning loss on unlabeled data. $\alpha$ is a hyperparameter to balance the learning of labeled and unlabeled data. In this paper, we utilize the self-labeling loss used in Wen et al. (2022) and detail it in the Appendix B.

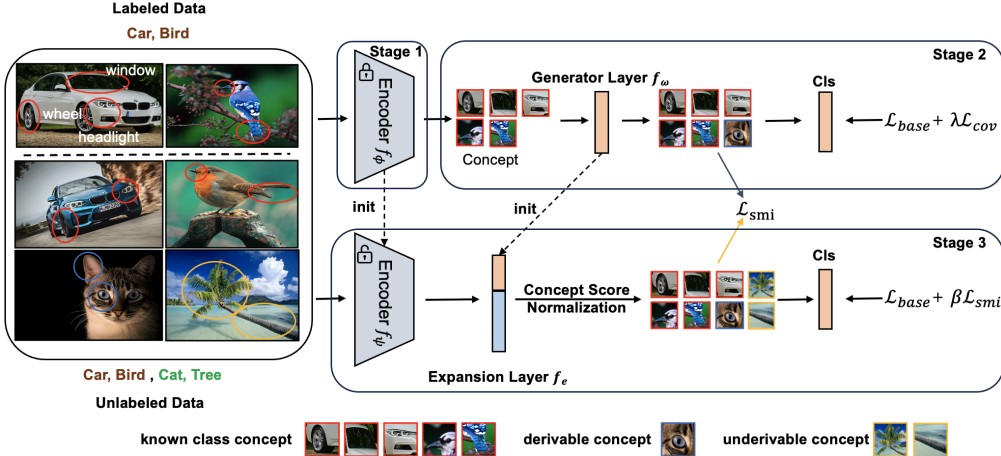

Figure 2: The overview of our novel ConceptGCD learning framework. Car, Bird, and Cat, Tree represent known and novel classes, respectively. 'Cls' denotes the classifier, and the circle in each image (left) represents the concepts present in the corresponding data. Our framework consists of three training stages. In the first stage (top left), we fine-tune an encoder using labeled known class data to learn known class concepts. In the second stage(top right), we train a generator layer (GL) that can derive concepts from known class concepts. The final stage (bottom) introduces an expansion layer (EL), which builds upon the GL by increasing its dimensionality and incorporating a concept score normalization technique. Both the encoder and the EL are concurrently trained to learn novel concepts while preserving previously learned concepts, guided by the loss function $\mathcal{L}_{smi}$. The concept shown above is for understanding, and the learned concepts are visualized in Appendix M.

## 3.2 MOTIVATIONS AND METHOD OVERVIEW

One of the main challenges in GCD is to transfer knowledge from known classes to novel classes. To tackle it, most existing methods (Vaze et al., 2022a; Wen et al., 2022; Zhang et al., 2022; Sun & Li, 2022; Choi et al., 2024) focused on the formulation of the unsupervised loss term $\mathcal{L}_u$ and establish a shared representation space for knowledge transfer from known to novel classes. However, this knowledge transfer is easily influenced by the noisy learning of unlabeled data. This may lead to the ineffective utilization of known class knowledge, as demonstrated in Tab. 1.

To better analyze this issue, following (Zeiler & Fergus, 2014; Allen-Zhu & Li, 2020a;b; Erhan et al., 2009; Nguyen et al., 2016), we introduce concepts as a way to represent knowledge. Specifically, we posit that each class has a certain concept set and neural networks inherently learn concepts throughout their training process. These learned concepts play a crucial role in the neural network's final classification. We define a concept $c$ as the input that maximizes the value of the corresponding neuron in the neural network. Consequently, the neuron's output for a given data represents the score assigned to the corresponding concept. Considering that the feature space may offer more capacity than the label space, we opt for the feature space as our chosen concept representation space. Then for an $n$-dimension feature space, the model will learn $n$ concepts $C = \{c_1, c_2, ..., c_n\}$. Additionally, we provide a visualization of these concepts in the Appendix M.

In the GCD problem, known class concepts $C^k = \{c_1^k, c_2^k, ..., c_l^k\}$ can be obtained by simply training a model on labeled data. However, acquiring full novel class concepts $C^u = \{c_1^u, c_2^u, ..., c_{l'}^u\}$ becomes challenging due to the absence of label information. Nevertheless, the semantic relationship between novel and known classes in GCD problems suggests that some novel class concepts are linked to known class concepts. Therefore, we believe these novel class concepts can be generated from known class concepts: Given $C^k$ and $C^u$, $\exists$ function $g$ and a subset $C^g \subseteq C^u$, s.t $C^g = g(C^k)$, leading to a classification of all class concepts $C = C^k \bigcup C^u$ into two groups: those derivable from known class concepts and those that are not. The strong performance observed when directly using known class concepts— a subset of the derivable concepts—as shown in Tab. 1, highlights the importance of derivable concepts. It also indicates that current methods struggle to effectively capture these derivable concepts, as they attempt to learn both derivable and underivable concepts simultaneously

using the same encoder. Consequently, the known class knowledge is compromised by noise from label uncertainty, resulting in low-quality derivable concepts.

Drawing from these, as illustrated in Fig. 2, we propose our novel concepts learning framework, named ConceptGCD, consisting of three core stages: 1) *Learn known class concepts*. This stage is dedicated to learning known class concepts by a pre-trained known class model. 2) *Generate derivable concepts*. This stage composes known class concepts to generate derivable concepts for novel classes. 3) *Learn underivable concepts*. This stage involves learning new concepts that cannot be derived from the known class concepts while retaining the generated concepts. In the subsequent sections, we will provide a comprehensive explanation of each stage in our novel framework.

### 3.3 CONCEPTGCD: A CONCEPT-DRIVEN APPROACH

As discussed above, our ConceptGCD framework has three key stages. In this section, we first introduce our model's architecture and then detail each stage.

**Architecture.**    To gain a powerful feature representation space, our encoder is a self-supervised pre-trained vision transform. Specifically, we employ the DINO pre-trained ViT-B/16 (Caron et al., 2021) and DINOv2 pre-trained ViT-B/14 (Oquab et al., 2023) as our encoders. As illustrated in Fig. 2, our method consists of two branches with similar structures. In the upper branch, we utilize a frozen pre-trained known-class encoder $f_\phi$ to capture $l$ known class concepts. Following $f_\phi$, we append a $l \times m$ linear layer and a ReLU layer, serving as our generator layer $f_\omega$ to produce $m$ derivable concepts. In the lower branch, we learn encoder $f_\psi$ initialized from $f_\phi$ and append a $l \times n$ linear layer and a ReLU layer, functioning as our expansion layer $f_e$ to learn $n - m$ underivable concepts while preserving $m$ derivable concepts. In both two branches, we append a linear layer after $f_\omega$ and $f_e$ serving as our classifier. Notably, in our final model, we retain only the lower branch.

**Learn Known Class Concepts.**    Building upon the idea that neural networks inherently learn class concepts during training (Zeiler & Fergus, 2014; Allen-Zhu & Li, 2020a;b), we train a model exclusively on the known class data to learn known class concepts. Specifically, we utilize the feature space of the encoder $f_\phi$ from the pre-trained known-class model as a means to represent the known class concepts. This choice is supported by our findings in Tab. 1, where we observed that this feature space effectively categorizes known class data and clusters novel class data. Furthermore, since the subsequent concepts are generated from the concepts learned at this stage, we aim to maximize the concept space here to ensure the high-quality generation of new concepts. Therefore, it is essential that the concepts learned in this stage are as independent as possible. To achieve this, we draw inspiration from Zbontar et al. (2021); Bardes et al. (2021) and apply a covariance loss. This loss function minimizes the covariance between the responses of the concepts, promoting their independence. Formally, we define $\mathbf{Z} = [\mathbf{z}_1, \mathbf{z}_2, ..., \mathbf{z}_B] = f_\phi(X)$, which consists of $B$ vectors of dimension $l$, where $l$ represents both the dimension of the encoder's feature space and the number of known class concepts, and $B$ is the batch size. The covariance matrix of $Z$ is given by:

$$C(Z) = \frac{1}{B-1} \sum_{i=1}^{B} \left( \mathbf{z}_i - \bar{\mathbf{z}} \right) \left( \mathbf{z}_i - \bar{\mathbf{z}} \right)^T, \text{ where } \bar{\mathbf{z}} = \frac{1}{B} \sum_{i=1}^{B} \mathbf{z}_i \tag{2}$$

When the batch size $B$ is sufficiently large, $C_{i,j}$ approximates the covariance between concept $i$ and concept $j$. The concept covariance loss can then be defined as:

$$\mathcal{L}_{cov} = \frac{1}{l(l-1)} \sum_{i \neq j} [C(Z)]_{i,j}^2 \tag{3}$$

The overall loss in this stage is:

$$\mathcal{L}_{1st} = \mathcal{L}_s + \lambda \mathcal{L}_{cov} \tag{4}$$

where $\mathcal{L}_s$ is the standard cross-entropy loss (the supervised loss term in $\mathcal{L}_{base}$), and $\lambda$ is a hyper-parameter controlling the weight of the covariance loss. In all settings, we simply set $\lambda$ to 1. By introducing this concept covariance loss, the model is encouraged to learn a more diverse set of known class concepts, thereby providing a larger concept space for the subsequent generation step.

**Generate Derivable Concepts.** Under the premise that some novel class concepts can be derived from known class concepts, we propose a method for generating novel class concepts once the known class concepts have been acquired. Our approach involves introducing generator layer $f_\omega$, which comprises a linear and a ReLU layer after the frozen known class pre-trained encoder $f_\phi$, and training $f_\omega$ on both labeled and unlabeled data using $\mathcal{L}_{2nd} = \mathcal{L}_{base} + \lambda \mathcal{L}_{cov}$. The concept covariance loss, $\mathcal{L}_{cov}$, is applied on all dimensions to encourage the model to learn a more diverse and expansive concept space, which may facilitate learning novel classes. The generator layer composes known class concept scores to determine the scores on the generated new concepts. This straightforward design and the frozen encoder preserve essential original known class concepts while generating new concepts for novel classes in a low-noise environment. For convenience, we denote the composite module as $f_c$ and represent its output as $\mathbf{v} = f_c(x)$, where $f_c = f_\omega \cdot f_\phi$.

**Learn Underivable Concepts.** Because the original encoder $f_\phi$ is exclusively trained on labeled known class data, the model's capacity to learn underivable novel concepts is constrained when $f_\phi$ is utilized as the final encoder. To address this limitation, we train a new encoder $f_\psi$, which is initialized from $f_\omega$, on both labeled and unlabeled data, thereby enabling the acquisition of underivable concepts. Additionally, we introduce an expansion layer $f_e$, which replaces and expands the previous generator layer $f_\omega$. Specifically, our expansion layer $f_e$ also comprises a linear and a ReLU layer. While $f_e$ retains a structure similar to $f_\omega$, it differs in that the output dimension is increased from $m$ to $n$ to learn $n - m$ new concepts. For convenience, we denote the final model as $f_\theta = f_e \cdot f_\psi$.

During the new model training, due to $f_\psi$ trained differently compared to $f_\phi$, it is necessary to ensure that the model retains generated concepts from the second stage. Since the value on each dimension of the feature represents the score on a specific concept, the $m$ generated concepts can be retained by ensuring that the model exhibits a consistent response on the corresponding dimensions of the feature. Inspired by (Tian et al., 2019), we adopt contrastive learning to achieve that. In detail, we take the two representations of the same unlabeled data $x_i$ in two representation spaces, $\mathbf{u}^i = f_\theta(x_i)$, $\mathbf{v}^i = f_c(x_i)$ as a positive pair meanwhile we take $\mathbf{u}^i$ and the generated features from the negative sample generator as negative pairs. Therefore, the knowledge transfer constraint term in the top $m$ dimensions is formulated as:

$$\mathcal{L}_{smi} = -\frac{1}{B} \sum_{i=1}^{B} \log \frac{e^{(\mathbf{u}^i_{1:m})^\top \mathbf{v}^i_{1:m}/\tau}}{e^{(\mathbf{u}^i_{1:m})^\top \mathbf{v}^i_{1:m}/\tau} + \sum_{\mathbf{z} \in \mathcal{N}} e^{(\mathbf{u}^i_{1:m})^\top \mathbf{z}_{1:m}/\tau}}, \tag{5}$$

where $\tau$ is the hyperparameter of temperature, $\mathcal{N}$ is the set of the negative samples in memory and $\mathbf{u}^i_{1:m}$ denotes the first $m$ values of the vector $\mathbf{u}^i$. It is important to note that all vectors in this equation are normalized, although we omit the normalization step for simplicity. With this loss, the new model will have consistent responses on the top $m$ dimensions, thereby maintaining generated concepts.

At this stage, we expand the dimension of the linear layer from $m$ to $n$, allowing the model to theoretically learn $n - m$ new concepts. However, our experimental findings (refer to Appendix E) reveal that the activation related to these new concepts is markedly weak, suggesting that the model predominantly relies on previously generated concepts and struggles to learn underivable concepts. It also indicates that these newly acquired concepts may largely represent noise, which could undermine the known class knowledge in the model. To mitigate this issue, we propose a Concept Score Normalization technique in the expansion layer. This operation normalizes the feature vector $\mathbf{u} = f_\theta(x)$ as $\mathbf{u}'$:

$$\mathbf{u}'_{1:m} = \sqrt{m} \frac{\mathbf{u}_{1:m}}{||\mathbf{u}_{1:m}||}, \quad \mathbf{u}'_{m+1:n} = \sqrt{n - m} \frac{\mathbf{u}_{m+1:n}}{||\mathbf{u}_{m+1:n}||} \tag{6}$$

where $|| \cdot ||$ denotes the L2-norm. By incorporating this normalization, the model is learned in a more balanced manner and is encouraged to learn more important concepts.

The overall loss in third stage is:

$$\mathcal{L}_{3rd} = \mathcal{L}_{base} + \beta \mathcal{L}_{smi} \tag{7}$$

where $\beta$ is a hyperparameter that controls the strength of knowledge transfer between the stage two model and the stage three model. Notably, at this stage, we do not apply the concept covariance loss $\mathcal{L}_{cov}$ because the $\mathcal{L}_{smi}$ implicitly enforces that the model retains the concept-independent property developed in the second stage of training. A comprehensive analysis of this decision and its implications will be provided in Appendix D.

Finally, the concepts of the new model are comprised of the generated concepts and the brand-new concepts. This composite structure can ensure that the model uses known class knowledge effectively while retaining the ability to learn novel concepts independent of known classes.

### 3.4 LEARNING STRATEGY

We adopt a three-stage learning strategy to learn our framework. The first stage involves training the encoder $f_\phi$ on labeled known class data using $\mathcal{L}_{1st}$ to get known class concepts. In the second stage, to learn the generator layer $f_\omega$, we fix $f_\phi$, utilize the feature after the generator layer to perform classification and clustering, and adopt $\mathcal{L}_{2nd}$ to learn labeled and unlabeled data. In the third stage, we learn the joint representation space ($f_\theta$) and cosine classifier by $\mathcal{L}_{3rd}$ to retain generated concepts and learn new underivable concepts.

In summary, our novel concept learning framework enables a better utilization of known class knowledge. This ensures that known class knowledge not only persists but is also employed to compose novel class concepts. Furthermore, the three-stage learning process guarantees that the model maximizes the utilization of known class knowledge while retaining the capability to acquire novel class knowledge independent of known classes.

## 4 EXPERIMENTS

### 4.1 EXPERIMENTAL SETUP

**Benchmark** To validate the effectiveness of our method, we follow Vaze et al. (2022a) and conduct experiments on various datasets, including generic datasets such as CIFAR100 (Krizhevsky et al., 2009) and ImageNet100 (Deng et al., 2009), as well as the Semantic Shift Benchmark (Vaze et al., 2022b), namely CUB (Wah et al., 2011), Stanford Cars (Krause et al., 2013), FGVC-Aircraft (Maji et al., 2013), and imbalanced Herbarium19 (Tan et al., 2019) dataset.

**Evaluation protocol.** Similar to Vaze et al. (2022a), we evaluate the model on unlabeled datasets with clustering accuracy. Specifically, we first employ the Hungarian matching algorithm to obtain the best matching between cluster and ground truth, and then we report the performance separately on known, novel, and all classes.

**Implementation details.** We adopt the DINO (Caron et al., 2021) pre-trained ViT-B/16 (Dosovitskiy et al., 2020) and DINOv2 (Oquab et al., 2023) pre-trained ViT-B/14 as our backbone. For both backbones, we only finetune the last block of ViT. The generator and expansion layers each consist of a linear layer followed by a ReLU activation. Specifically, for the generator layer, we set $m = 2l$, resulting in a linear layer dimension of $768 \times 1536$. For the expansion layer, we set $n = 10l$, leading to a dimension of $768 \times 7680$. We provide a detailed analysis of these design choices in the Appendix. In the first stage, we train our model by 100 epochs on labeled data. In the second stage, we train our model by 100 epochs on all data. We adopt the SGD optimizer with a momentum of 0.9, a weight decay of $5 \times 10^{-5}$, and an initial learning rate of 1.0, which reduces to $1e - 4$ at 100 epochs using a cosine annealing schedule. The batch size is 128 and the data augmentation is the same as Vaze et al. (2022a). Standard hyperparameters are set in convention as follows: $\alpha = 0.35, \epsilon = 1$ as in Vaze et al. (2022a); Xu et al. (2022), $\tau = 0.1$ with initial $\tau' = 0.07$ warmed up to 0.04 over the first 30 epochs using a cosine schedule as in Caron et al. (2021). For the hyperparameters $\beta$ and $\lambda$ that we introduce, we set $\beta$ to 0.1 and $\lambda$ to 1.0 for all datasets. We validate those two hyperparameters in the Appendix. All the experiments are conducted on a single NVIDIA TITAN RTX.

### 4.2 COMPARISON WITH STATE-OF-THE-ART METHODS

**Main Results.** Tab. 3 and Tab. 4 show the performance comparison of our model against current state-of-the-art (SOTA) methods with DINO (Caron et al., 2021) and DINOv2 (Oquab et al., 2023), especially on novel classes. For instance, on the CIFAR100-80 dataset, our approach achieves a notable increase of 4.6% in accuracy over CMS (Choi et al., 2024) for novel classes. For the CUB dataset, while our overall performance is comparable to that of InfoSieve (Rastegar et al., 2023), we achieve a 1.3% improvement in the novel classes. When compared with SPTNet (Wang et al., 2024a) on the Stanford Cars dataset, our model demonstrates significant gains, with an 11.1% improvement for all classes and a 15.3% increase for novel classes. Similarly, on the FGVC-Aircraft dataset, our

Table 3: Comparison with state-of-the-art methods."crNCD*" means we re-implement crNCD (Gu et al., 2023) in the GCD setting.

| Method | CIFAR100-80 | | | ImageNet100-50 | | | CUB | | | StanfordCars | | | FGVC-Aircraft | | |
|---|---|---|---|---|---|---|---|---|---|---|---|---|---|---|---|
| | All | Known | Novel | All | Known | Novel | All | Known | Novel | All | Known | Novel | All | Known | Novel |
| K-means | 52.0 | 52.2 | 50.8 | 72.7 | 75.5 | 71.3 | 34.3 | 38.9 | 32.1 | 12.8 | 10.6 | 13.8 | 16.0 | 14.4 | 16.8 |
| RS+ (Zhao & Han, 2021) | 58.2 | 77.6 | 19.3 | 37.1 | 61.1 | 24.8 | 33.3 | 51.6 | 24.2 | 28.3 | 61.8 | 12.1 | 26.9 | 36.4 | 22.2 |
| UNO (Fini et al., 2021) | 69.5 | 80.6 | 47.2 | 70.3 | 95.0 | 57.9 | 35.1 | 49.0 | 28.1 | 35.5 | 70.5 | 18.6 | 40.3 | 56.4 | 32.2 |
| ORCA (Cao et al., 2022) | 69.0 | 77.4 | 52.0 | 73.5 | 92.6 | 63.9 | 35.3 | 45.6 | 30.2 | 23.5 | 50.1 | 10.7 | 22.0 | 31.8 | 17.1 |
| GCD (Vaze et al., 2022a) | 70.8 | 77.6 | 57.0 | 74.1 | 89.8 | 66.3 | 51.3 | 56.6 | 48.7 | 39.0 | 57.6 | 29.9 | 45.0 | 41.1 | 46.9 |
| PromptCAL (Zhang et al., 2022) | 81.2 | 84.2 | 75.3 | 83.1 | 92.7 | 78.3 | 62.9 | 64.4 | 62.1 | 50.2 | 70.1 | 40.6 | 52.2 | 52.2 | 52.3 |
| DCCL (Pu et al., 2023) | 75.3 | 76.8 | 70.2 | 80.5 | 90.5 | 76.2 | 63.5 | 60.8 | 64.9 | 43.1 | 55.7 | 36.2 | - | - | - |
| SimGCD (Wen et al., 2022) | 78.1 | 77.6 | 78.0 | 82.4 | 90.7 | 78.3 | 60.3 | 65.6 | 57.7 | 46.8 | 64.9 | 38.0 | 48.8 | 51.0 | 47.8 |
| crNCD* (Gu et al., 2023) | 80.4 | 85.3 | 70.6 | 81.7 | 91.3 | 76.9 | 64.1 | 75.2 | 58.6 | 54.8 | 76.5 | 44.3 | 53.1 | 57.0 | 51.3 |
| μGCD (Vaze et al., 2024) | - | - | - | - | - | - | 65.7 | 68.0 | 64.6 | 56.5 | 68.1 | 50.9 | 53.8 | 55.4 | 53.0 |
| InfoSieve (Rastegar et al., 2023) | 78.3 | 82.2 | 70.5 | 80.5 | 93.8 | 73.8 | **69.4** | **77.9** | 65.2 | 55.7 | 74.8 | 46.4 | 56.3 | **63.7** | 52.5 |
| LegoGCD (Cao et al., 2024) | 81.8 | 81.4 | 98.5 | 86.3 | 94.5 | 82.1 | 63.8 | 71.9 | 59.8 | 57.3 | 75.7 | 48.4 | 55.0 | 61.5 | 51.7 |
| CMS (Choi et al., 2024) | 82.3 | **85.7** | 75.5 | 84.7 | **95.6** | 79.2 | 68.2 | 76.5 | 64.0 | 56.9 | 76.1 | 47.6 | 56.0 | 63.4 | 52.3 |
| SPTNet (Wang et al., 2024a) | 81.3 | 84.3 | 75.6 | 85.4 | 93.2 | 81.4 | 65.8 | 68.8 | 65.1 | 59.0 | 79.2 | 49.3 | 59.3 | 61.8 | 58.1 |
| ConceptGCD (Ours) | **82.8** | 84.1 | **80.1** | **86.3** | 93.3 | **82.8** | **69.4** | 75.4 | **66.5** | **70.1** | **81.6** | **64.6** | **60.5** | 59.2 | **61.1** |

Table 4: Herbarium19

| Method | Herbarium19 | | |
|---|---|---|---|
| | All | Known | Novel |
| GCD (Vaze et al., 2022a) | 35.4 | 51.0 | 27.0 |
| SimGCD (Wen et al., 2022) | 44.0 | 58.0 | 36.4 |
| PromptCAL (Zhang et al., 2022) | 37.0 | 52.0 | 28.9 |
| InfoSieve (Rastegar et al., 2023) | 41.0 | 55.4 | 33.2 |
| SPTNet (Wang et al., 2024a) | 43.4 | **58.7** | 35.2 |
| ConceptGCD (Ours) | **45.5** | 56.2 | **39.7** |

Table 5: Results with DINOV2 Backbone

| Method | CUB | | | Stanford Cars | | | FGVC-Aircraft | | |
|---|---|---|---|---|---|---|---|---|---|
| | All | Known | Novel | All | Known | Novel | All | Known | Novel |
| Kmeans | 67.6 | 60.6 | 71.1 | 29.4 | 24.5 | 31.8 | 18.9 | 16.9 | 19.9 |
| GCD (Vaze et al., 2022a) | 71.9 | 71.2 | 72.3 | 65.7 | 67.8 | 64.7 | 55.4 | 47.9 | 59.2 |
| SimGCD (Wen et al., 2022) | 71.5 | 78.1 | 68.3 | 71.5 | 81.9 | 66.6 | 63.9 | 69.9 | 60.9 |
| μGCD (Vaze et al., 2024) | 74.0 | 75.9 | 73.1 | 76.1 | **91.0** | 68.9 | 66.3 | 68.7 | 65.1 |
| ConceptGCD (Ours) | **76.0** | **80.7** | **73.6** | **80.4** | 88.6 | **76.4** | **71.1** | **71.1** | **71.2** |

method achieves gains of 1.2% and 3.0% for all classes and novel classes, respectively, over SPTNet. Lastly, on the Herbarium19 dataset (Tab. 4), our method obtains significant gains over SPTNet: 2.1% and 4.5% on All and Novel metrics, respectively. These superior results demonstrate the efficacy of our proposed method.

**Results with DINOv2 backbone.** We follow μGCD (Vaze et al., 2024) and conduct experiments utilizing the DINOv2 backbone (ViT-B14). As illustrated in Tab. 5, our method yields significant improvements on the CUB dataset (4.8% for Known), Stanford Cars dataset (7.5% for Novel), and the Aircraft dataset (6.1% for Novel). These improvements further demonstrate the effectiveness of our method over different backbones.

### 4.3 ABLATION STUDY

In this section, we provide analysis of our method through multiple perspectives. We start with an ablation study to examine the contribution of each component in our approach. Next, to gain visual insights, we visualize the representation space of different models, including the pre-trained model, generator layer, and our final model. Meanwhile, we present our model in a more realistic situation where the number of clusters is unknown. These analyses provide a comprehensive understanding of our approach and its effectiveness in various scenarios.

**Baseline.** We first train the backbone on known class data. Then we freeze the backbone and learn a classifier head on both known and novel class data using $\mathcal{L}_{base}$ as defined in Eq. 1.

**Component analysis.** In Tab. 6, we conduct an ablation study to assess the effectiveness of four key components in our model: Generator Layer (GL), Concept Covariance Loss (1stCov, 2ndCov), Contrastive Loss (CL), and Concept Score Normalization (CSN). Corresponding to our conceptual framework, the "baseline" model (first row) utilizes only known class concepts. The second through fourth rows represent models that employ solely derivable concepts, while the fifth and sixth rows describe models that utilize derivable and underivable concepts. Our findings reveal that integrating a straightforward GL into a fixed model pre-trained on known classes with a simple loss ($\mathcal{L}_{base}$ in Eq. 1) can already match or surpass state-of-the-art outcomes on fine-grained datasets and yield commendable performance on coarse-grained datasets. Furthermore, incorporating the Concept Covariance Loss in stage 1 or stage 2 significantly enhances performance across almost all datasets, particularly with novel classes. This improvement is especially pronounced when applying the Concept Covariance Loss in stage 1, demonstrating its effectiveness in learning independent concepts. Additionally, CL improves performance on novel classes but slightly reduces performance on known classes. This may be due to the model learning noise that compromises the known class knowledge. More analyses are presented in Appendix E. The CSN addresses this issue and further boosts

Table 6: Ablation study. 'GL' stands for Generator Layer; '1stCov' and '2ndCov' represent the Concept Covariance Loss in the first and second stages, respectively; 'CL' denotes Contrastive Loss; and 'CSN' refers to the Concept Score Normalization. The gray shading indicates the performance metrics for the second stage model, while the white shading reflects the final model performance.

| GL | 1stCov | 2ndCov | CL | CSN | CIFAR100 | | | CUB | | | Stanford Cars | | | FGVC-Aircraft | | |
|---|---|---|---|---|---|---|---|---|---|---|---|---|---|---|---|---|
| | | | | | All | Known | Novel | All | Known | Novel | All | Known | Novel | All | Known | Novel |
| | | | | | 69.2 | 84.0 | 39.6 | 67.3 | 76.5 | 62.7 | 52.5 | 75.2 | 41.5 | 48.0 | 59.0 | 42.5 |
| ✓ | | | | | 79.1 | 83.9 | 69.5 | 68.6 | 76.7 | 64.5 | 60.7 | 77.0 | 52.8 | 57.3 | 60.6 | 55.6 |
| ✓ | ✓ | | | | 80.6 | 84.1 | 73.6 | 67.4 | 74.0 | 64.1 | 68.5 | 81.0 | 62.4 | 59.2 | 59.6 | 59.0 |
| ✓ | ✓ | ✓ | | | 81.9 | 84.5 | 76.7 | 68.4 | 74.0 | 65.6 | 70.0 | 81.4 | 64.5 | 60.0 | 59.0 | 60.6 |
| ✓ | ✓ | ✓ | ✓ | | 82.4 | 84.3 | 78.7 | 68.5 | 72.4 | 66.6 | 69.9 | 81.3 | 64.4 | 59.3 | 56.0 | 60.9 |
| ✓ | ✓ | ✓ | ✓ | ✓ | 82.8 | 84.1 | 80.1 | 69.4 | 75.4 | 66.5 | 70.1 | 81.6 | 64.6 | 60.5 | 59.2 | 61.1 |

Table 7: Unknown $N^n$. "*" denotes our method with known $N^n$; others treat $N^n$ as unknown.

| Method | CUB | | | Stanford Cars | | | FGVC-Aircraft | | |
|---|---|---|---|---|---|---|---|---|---|
| | All | Known | Novel | All | Known | Novel | All | Known | Novel |
| SimGCD(Wen et al., 2022) | 62.4 | 67.1 | 60.0 | 52.6 | 72.7 | 42.9 | 52.3 | 56.2 | 50.3 |
| ConceptGCD (Ours) | 68.5 | 72.9 | 66.2 | 69.3 | 80.8 | 63.8 | 59.9 | 57.6 | 61.1 |
| ConceptGCD (Ours)* | 69.4 | 75.4 | 66.5 | 70.1 | 81.6 | 64.6 | 60.5 | 59.2 | 61.1 |

Pre-trained Known-class Model  Generator Layer  Our Final Model

Figure 3: t-SNE visualization on CIFAR100-80. More visualization are in Appendix N.

performance across all datasets. In conclusion, these results endorse the utility of each individual component, collectively reinforcing the integrity of our overall model design.

**t-SNE visualization.** Fig. 9 offers a visualization of our model representation spaces in each stage using t-SNE. The visualization demonstrates a remarkable transformation of the model's representation space—transitioning from a dispersed and chaotic arrangement in the pre-trained known-class model to a denser and more orderly structure after interfacing with the generator layer. This transformation is consistent with the objective of our generator layer, which is devised to facilitate the generation of concepts associated with novel classes. Subsequently, the final model more effectively clusters categories into compact groups, particularly for novel classes. This is consistent with our design intent, which aimed to enhance the model's ability to learn novel class concepts.

**The number of clusters $N^n$ is unknown.** The experiments presented so far assume that the number of clusters is known a priori, which is often unrealistic in practice. To address this limitation, we employ the method proposed in (Vaze et al., 2022a) to infer the number of classes for each dataset. Specifically, we consider FGVC-Aircraft to have 108 classes, CUB to have 231 classes, and Stanford Cars to have 230 classes. We then conduct experiments using these estimated class numbers. As Tab. 7 show, our method significantly improves over the baseline for both known and novel classes. Furthermore, the performance only shows a minor decline compared to scenarios where $N^n$ is known. These findings underscore the robustness of our approach in realistic settings.

## 5 DISCUSSION

This paper presents a novel and straightforward concept learning framework for generalized category discovery, aiming to enhance the efficient utilization of known class knowledge while preserving the model's ability to learn new novel class knowledge independently from known classes. The framework consists of three key steps: 1) Learning known class concepts: train a model on known class data with a covariance-augmented loss to acquire known class concepts; 2) Generating derivable concepts: utilize a generator layer to learn derivable concepts; and 3) Learning underivable concepts:

expand the generator layer and utilize a contrastive loss and a concept score normalization technique, ensuring that the model retains generated concepts while learning new independent concepts in a balanced manner. Extensive evaluations demonstrate the remarkable superiority of our approach compared to existing methods in the field. Furthermore, our novel concept learning framework introduces a fresh perspective for the utilization of known class knowledge in generalized category discovery while retaining the ability to learn new knowledge. The findings of this study can serve as a strong baseline for future work and hold promise for addressing the critical challenge of effectively transferring knowledge from known to novel classes in GCD.

**Limitations** Although our method is novel and has achieved remarkable results over existing approaches, it has some limitations: 1) Multiple training stages: Our method involves three stages of training. While each stage is simple and requires training only a small number of parameters, the process is still a little complex. Future methods could aim to simplify this multi-stage approach.; 2) Less flexible concept learning strategy: During the third stage, when learning new concepts, our method indiscriminately retains all concepts learned in the second stage. However, as demonstrated by Zhao et al. (2024), not all known class data is useful, and therefore, we believe not all learned concepts are beneficial. A selective mechanism may be needed to dynamically filter out less useful concepts while adding new ones, rather than retaining all existing concepts; 3) Limited theoretical interpretability: As shown in the Appendix M, while our concepts possess some degree of interpretability, more theoretical analyses are needed to fully explain these concepts thereby enhancing our understanding of the generalized category discovery. We hope that future research will introduce more advanced methods to address the limitations mentioned above.

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

# A DATASETS

We conduct experiments on widely-used datasets such as CIFAR100 (Krizhevsky et al., 2009) and ImageNet100 (Deng et al., 2009), as well as the recently proposed Semantic Shift Benchmark (Vaze et al., 2022b), namely CUB (Wah et al., 2011), Stanford Cars (Scars) Krause et al. (2013), FGVC-Aircraft (Maji et al., 2013) and Herbarium-19 (Tan et al., 2019). The details of the split are as follows:

Table 8: The detail of datasets.

| Dataset | Labeled $\mathcal{D}^l$ | | Unlabeled $\mathcal{D}^u$ | |
|---|---|---|---|---|
| | #Image | #Class | #Image | #Class |
| CIFAR100 | 20K | 80 | 30k | 100 |
| ImageNet100 | 31.9K | 50 | 95.3K | 50 |
| CUB | 1.5K | 100 | 4.5K | 200 |
| Stanford Cars | 2.0K | 98 | 6.1K | 196 |
| FGVC-Aircraft | 1.7K | 50 | 5.0K | 100 |
| Herbarium-19 | 8.9K | 341 | 25.4K | 683 |

# B THE DETAILS OF $\mathcal{L}_u$

In this paper, we adopt the self-labeling loss (Caron et al., 2021; Wen et al., 2022) as our $\mathcal{L}_u$. Specifically, for each unlabeled data point $x_i$, we generate two views $x_i^{v_1}$ and $x_i^{v_2}$ through random data augmentation. These views are then fed into the ViT (Dosovitskiy et al., 2020) encoder and cosine classifier ($h$), resulting in two predictions $\mathbf{y}_i^{v_1} = h(f_\theta(x_i^{v_1}))$ and $\mathbf{y}_i^{v_2} = h(f_\theta(x_i^{v_2}))$, $\mathbf{y}_i^{v_1}, \mathbf{y}_i^{v_2} \in \mathbb{R}^{C^k + C^n}$. As we expect the model to produce consistent predictions for both views, we employ $\mathbf{y}_i^{v_2}$ to generate a pseudo label for supervising $\mathbf{y}_i^{v_1}$. The probability prediction and its pseudo label are denoted as:

$$\mathbf{p}_i^{v_1} = \text{Softmax}(\mathbf{y}_i^{v_1}/\tau), \quad \mathbf{q}_i^{v_2} = \text{Softmax}(\mathbf{y}_i^{v_2}/\tau') \tag{8}$$

Here, $\tau, \tau'$ represents the temperature coefficients that control the sharpness of the prediction and pseudo label, respectively. Similarly, we employ the generated pseudo-label $\mathbf{q}_i^{v_1}$, based on $\mathbf{y}_i^{v_1}$, to supervise $\mathbf{y}_i^{v_2}$. However, self-labeling approaches may result in a degenerate solution where all novel classes are clustered into a single class (Caron et al., 2018). To mitigate this issue, we introduce an additional constraint on cluster size. Thus, the loss function can be defined as follows:

$$\mathcal{L}_u = \frac{1}{2|\mathcal{D}^u|} \sum_{i=1}^{|\mathcal{D}^u|} [l(\mathbf{p}_i^{v_1}, \text{SG}(\mathbf{q}_i^{v_2})) + l(\mathbf{p}_i^{v_2}, \text{SG}(\mathbf{q}_i^{v_1}))] + \epsilon \mathbf{H}(\frac{1}{2|\mathcal{D}^u|} \sum_{i=1}^{|\mathcal{D}^u|} \mathbf{p}_i^{v_1} + \mathbf{p}_i^{v_2}) \tag{9}$$

Here, $l(\mathbf{p}, \mathbf{q}) = -\mathbf{q} \log \mathbf{p}$ represents the standard cross-entropy loss, and SG denotes the "stop gradient" operation. The entropy regularizer $\mathbf{H}$ enforces cluster size to be uniform thus alleviating the degenerate solution issue. The parameter $\epsilon$ represents the weight of the regularize.

# C THE DETAILS OF $\mathcal{N}$

Similar to traditional contrastive learning (He et al., 2020), we treat all other instances as negative samples without any hard example mining strategy. In detail, the memory buffer contains 2048 negative samples.

# D ANALYSIS OF THE CONCEPT COVARIANCE LOSS IN THE THIRD STAGE

In Tab. 9, we analyze the impact of incorporating $\mathcal{L}_{cov}$ during the third stage of model training. Specifically, we modify the loss function in the third stage from $\mathcal{L}_{3rd}$ to $\mathcal{L}_{base} + \beta\mathcal{L}_{smi} + \lambda\mathcal{L}_{cov}$ and train our model by this revised loss. As indicated in Tab. 9, $\mathcal{L}_{cov}$ is actually very small even when it is not used. This is likely because $\mathcal{L}_{smi}$ preserves the feature space structured in the second

Table 9: Performance of the final model and $\mathcal{L}_{cov}$ values with and without inclusion of $\mathcal{L}_{cov}$ in $\mathcal{L}_{3rd}$.

| 3rdCov | CIFAR100 | | | | CUB | | | | Stanford Cars | | | | FGVC-Aircraft | | | |
|---|---|---|---|---|---|---|---|---|---|---|---|---|---|---|---|---|
| | All | Known | Novel | $\mathcal{L}_{cov}$ | All | Known | Novel | $\mathcal{L}_{cov}$ | All | Known | Novel | $\mathcal{L}_{cov}$ | All | Known | Novel | $\mathcal{L}_{cov}$ |
| | 82.8 | 84.1 | 80.1 | 0.0007 | 69.4 | 75.4 | 66.5 | 0.0030 | 70.1 | 81.6 | 64.6 | 0.0003 | 60.5 | 59.2 | 61.1 | 0.0005 |
| ✓ | 82.7 | 83.6 | 80.9 | 0.0001 | 69.4 | 74.9 | 66.6 | 0.0031 | 70.3 | 81.7 | 64.8 | 0.0004 | 60.6 | 59.6 | 61.1 | 0.0006 |

Table 10: Values of $\|\mathbf{u}_{m:n}\|/\|\mathbf{u}\|$ without Concept Score Normalization (CSN). Here, $\mathbf{u} = f_\theta(x)$ as defined in Sec. 3.3. The notation $\|\cdot\|$ denotes the L2 norm.

| CSN | CIFAR100 | CUB | Stanford Cars | FGVC-Aircraft |
|---|---|---|---|---|
| ✗ | 0.44 | 0.24 | 0.13 | 0.09 |

Table 11: Performance of the Generator Layer with different depths.

| GL depth | CIFAR100 | | | CUB | | | Stanford Cars | | | FGVC-Aircraft | | |
|---|---|---|---|---|---|---|---|---|---|---|---|---|
| | All | Known | Novel | All | Known | Novel | All | Known | Novel | All | Known | Novel |
| 0 | 69.2 | 84.0 | 39.6 | 67.3 | 76.5 | 62.7 | 52.5 | 75.2 | 41.5 | 48.0 | 59.0 | 42.5 |
| 1 | 79.1 | 83.9 | 69.5 | 68.6 | 76.7 | 64.5 | 60.7 | 77.0 | 52.8 | 57.3 | 60.6 | 55.6 |
| 2 | 80.3 | 82.2 | 76.5 | 66.1 | 72.8 | 62.8 | 58.7 | 71.8 | 52.3 | 54.8 | 52.3 | 56.0 |
| 3 | 79.2 | 81.1 | 75.4 | 62.0 | 68.0 | 59.0 | 54.2 | 67.8 | 47.6 | 53.4 | 53.1 | 53.6 |

stage, allowing the third stage model to potentially inherit the concept independence property of the second stage model. Furthermore, the addition of $\mathcal{L}_{cov}$ to the third stage results in a negligible change in model performance. Consequently, for simplicity, we opted to exclude $\mathcal{L}_{cov}$ in the third stage's training process.

## E ANALYSIS OF THE CONCEPT SCORE NORMALIZATION

In the ablation study (Sec. 4.3), we demonstrated the importance of Concept Score Normalization (CSN) from the perspective of model performance. In this section, we further elucidate the significance of CSN from the perspective of the model features themselves. Tab. 10 presents the average value of $\|\mathbf{u}_{m:n}\|/\|\mathbf{u}\|$ across all data when CSN is not applied. We observe that this value is significantly low across almost all datasets, except for CIFAR100. This indicates that the model's learned concepts are rarely activated in the data, suggesting that these new concepts may be noise and may deteriorate the model's original known class knowledge.

To address this, we enlarge the influence of newly learned concepts on the model by Concept Score Normalization, thereby enabling the model to learn more useful concepts. As shown in Table 6, CSN significantly improves performance on novel classes in coarse-grained datasets, while in fine-grained datasets, it predominantly enhances performance on known classes. This disparity arises because known and novel classes are closely related in fine-grained datasets, making most concepts derivable from known class concepts. Consequently, there are few truly novel class concepts to learn in the third stage, resulting in only minor improvements for novel classes. Furthermore, because CSN helps the model preserve known class knowledge by reducing noisy concepts, model performance on known classes will be maintained and even improved on some datasets in the final stage.

## F ANALYSIS OF THE GENERATOR LAYER DEPTH

In our approach, we introduce a generator layer (GL) after the encoder of a model pre-trained on known classes. Here, we focus on demonstrating the importance of leveraging knowledge from known classes by training the GL with a typical loss (Wen et al., 2022), as defined in Eq. 1. Notably, this loss is only a portion of the total loss $\mathcal{L}_{2nd}$ ultimately used. Each unit within the GL consists of a linear layer followed by a ReLU activation function. To evaluate the impact of various configurations of the generator layer, we conduct experiments with varying depths of GL. Notably, GL with 0 depth implies training only the classifier head, which is also the baseline. The results, as shown in Tab. 11, reveal that a single generator layer attains superior performance on fine-grained datasets, even outperforming existing state-of-the-art methods. Notably, the two MLP layers design yields the most favorable outcome for the CIFAR100 dataset, suggesting that coarse-grained datasets might require additional flexibility to discover novel concepts. *These impressive outcomes underscore that models pretrained on known classes possess valuable knowledge for novel class discovery; however, current*

Table 12: Generator Layer performance with different numbers of output dimensions.

| GL dim | CIFAR100 | | | CUB | | | Stanford Cars | | | FGVC-Aircraft | | |
|---|---|---|---|---|---|---|---|---|---|---|---|---|
| | All | Known | Novel | All | Known | Novel | All | Known | Novel | All | Known | Novel |
| 768 | 80.4 | 84.6 | 72.0 | 68.5 | 73.5 | 66.1 | 68.7 | 81.4 | 62.6 | 60.0 | 58.8 | 60.6 |
| 1536 | 81.9 | 84.5 | 76.7 | 68.4 | 74.0 | 65.6 | 70.0 | 81.4 | 64.5 | 60.0 | 59.0 | 60.6 |
| 3072 | 81.6 | 84.5 | 76.0 | 68.8 | 74.3 | 66.0 | 69.3 | 79.9 | 64.1 | 59.7 | 58.7 | 60.1 |
| 7680 | 81.8 | 84.5 | 76.3 | 68.7 | 73.6 | 66.2 | 69.3 | 80.3 | 64.0 | 59.7 | 59.4 | 59.9 |

Table 13: Performance of our final model with various output dimensions of the Expansion Layer. $m$ is the output dimension of the generator layer, which is 1536 in our model.

| EL dim | CIFAR100 | | | CUB | | | Stanford Cars | | | FGVC-Aircraft | | |
|---|---|---|---|---|---|---|---|---|---|---|---|---|
| | All | Known | Novel | All | Known | Novel | All | Known | Novel | All | Known | Novel |
| $1.0m$ | 82.8 | 84.2 | 79.9 | 68.4 | 72.3 | 66.5 | 70.2 | 81.2 | 64.8 | 59.5 | 56.5 | 61.1 |
| $1.5m$ | 82.8 | 84.4 | 79.5 | 68.7 | 72.7 | 66.7 | 70.1 | 81.5 | 64.6 | 59.7 | 57.1 | 61.0 |
| $2.0m$ | 82.7 | 84.2 | 79.6 | 68.8 | 72.9 | 66.8 | 70.2 | 81.0 | 65.0 | 59.9 | 57.7 | 61.1 |
| $5.0m$ | 82.8 | 84.1 | 80.1 | 69.4 | 75.4 | 66.5 | 70.1 | 81.6 | 64.6 | 60.5 | 59.2 | 61.1 |
| $10.0m$ | 82.6 | 84.2 | 79.3 | 64.3 | 69.2 | 61.9 | 70.1 | 82.2 | 64.3 | 61.1 | 60.7 | 61.3 |
| $20.0m$ | 81.9 | 84.2 | 77.3 | 43.1 | 37.7 | 45.8 | 50.2 | 59.6 | 45.7 | 60.3 | 65.4 | 57.7 |

Table 14: Hyperparameter $\beta$ analysis on CUB.

| $\beta$ | 0 | 0.01 | 0.02 | 0.05 | 0.10 | 0.20 | 0.50 | 1.00 |
|---|---|---|---|---|---|---|---|---|
| All | 68.1 | 69.2 | 69.3 | 69.4 | 69.4 | 69.2 | 69.2 | 69.1 |
| Known | 69.5 | 73.6 | 74.0 | 74.4 | 75.4 | 74.5 | 75.1 | 75.0 |
| Novel | 67.4 | 67.1 | 67.0 | 66.9 | 66.5 | 66.6 | 66.2 | 66.2 |

Table 15: Hyperparameter $\lambda$ analysis on Stanford Cars.

| $\lambda$ | 0 | 0.1 | 0.2 | 0.5 | 1.0 | 2.0 | 5.0 | 10.0 |
|---|---|---|---|---|---|---|---|---|
| All | 60.7 | 65.6 | 66.1 | 68.8 | 70.0 | 70.2 | 69.3 | 68.8 |
| Known | 77.0 | 77.8 | 77.6 | 80.1 | 81.4 | 81.2 | 81.0 | 80.4 |
| Novel | 52.8 | 59.7 | 60.6 | 63.4 | 64.5 | 64.8 | 63.7 | 63.1 |

*methods in GCD may not fully leverage this potential.* Conversely, the three MLP layers design leads to a dip in results. This could be because the presence of excessive learning capacity permits noisy learning from the unlabeled data to detrimentally affect the learned representations. This observation further supports the finding that existing methods (Wen et al., 2022; Vaze et al., 2022a; 2024), which naively fine-tune the last block of the ViT, present diminished outcomes.

## G ANALYSIS OF THE GENERATOR LAYER DIMENSION

We conduct experiments on the generator layer with varying output dimensions, which determine the number of generated concepts and serve as the hyperparameter $m$ in our model. As shown in Tab. 12, when the GL dimension is set to 1536 the generator layer achieves satisfactory performance across all datasets. This table also demonstrates that our generator layer performs well over a wide range of GL dimensions, indicating the robustness of our model. Notably, in this experiment, we train the generator layer using $\mathcal{L}_{2nd}$, as defined in Sec. 3.3.

## H ANALYSIS OF THE EXPANSION LAYER DIMENSION

We investigate the effects of varying output dimensions in the expansion layer, which determine the total number of concepts and act as the hyperparameter $n$ in our model. Tab. 13 illustrates that when the EL dimension is set to $5m = 7680$, the final model achieves satisfactory performance across all datasets. This result further confirms that the expansion layer performs consistently well over a diverse range of EL dimensions, underlining our model's robustness.

## I HYPERPARAMETER $\beta$ ANALYSIS

We conduct the hyperparameter analysis of $\beta$ on CUB in Tab. 14. Our results indicate that the model maintains stable performance across a range of 0.01-0.5, indicating low sensitivity to $\beta$.

## J  HYPERPARAMETER $\lambda$ ANALYSIS

In our method, we simply set $\lambda$ to 1.0. Tab. 15 presents a hyperparameter analysis of $\lambda$, demonstrating that the model's performance remains stable within the range of 0.5 to 10.0. This stability indicates the model's robustness to variations in $\lambda$.

## K  MEAN AND VARIANCE ANALYSIS

We conducted all of our experiments three times, except for the ImageNet dataset due to the high computational and time costs. The low variance observed in our results underscores the reliability and stability of the method we have proposed.

| Dataset | All | Seen | Novel |
|---|---|---|---|
| CIFAR100 | $82.8 \pm 0.09$ | $84.1 \pm 0.12$ | $80.1 \pm 0.45$ |
| CUB | $69.4 \pm 0.61$ | $75.4 \pm 1.17$ | $66.5 \pm 0.33$ |
| StanfordCars | $70.1 \pm 0.52$ | $81.6 \pm 1.18$ | $64.6 \pm 1.25$ |
| Aircraft | $60.5 \pm 1.30$ | $59.2 \pm 0.36$ | $61.1 \pm 2.13$ |

Table 16: Mean and variance for ConceptGCD on Various Datasets

## L  EXPERIMENTS ON NAIVE RESNET18

| Dataset | CIFAR | | | ImageNet100 | | | CUB200 | | |
|---|---|---|---|---|---|---|---|---|---|
| | All | Known | Novel | All | Known | Novel | All | Known | Novel |
| SimGCD | 52.4 | 63.5 | 30.2 | 32.6 | 75.1 | 11.2 | 14.37 | 21.61 | 10.74 |
| Ours | 56.7 | 61.9 | 46.4 | 47.2 | 69.5 | 36.0 | 16.97 | 21.15 | 14.88 |

Table 17: Performance comparison on different datasets

To mitigate the influence of pre-trained models on our approach, we employ a ResNet18 architecture trained from scratch. Our method has demonstrated significant enhancements in performance, particularly in discovering novel classes across three distinct datasets. It is important to highlight that the results for SimGCD on the ImageNet100 dataset are the most optimal we have achieved to date.

## M  CONCEPT VISUALIZATION

In this section, we present the concept visualization in Fig. 4 and Fig. 5 using Grad-CAM(Selvaraju et al., 2017). As depicted in Fig. 4, the known class pre-trained model can successfully capture meaningful concepts such as "rabbit feet" (concept No. 226) in CIFAR100, "Least Auklet chest" (concept No. 137), and "Least Auklet belly" (concept No. 206) in CUB. Additionally, certain known class concepts, such as concept No. 412 in CIFAR100 and concept No. 597 in CUB, exhibit high scores on novel class data, suggesting that some of the known class concepts in the known class pre-trained model are related to novel classes, which aligns with our motivation.

Moreover, as shown in the middle part of Fig. 4, the generator layer can capture novel class concepts, such as "turtle neck" (concept No. 55) and "turtle head" (concept No. 1216) in CIFAR100, as well as "Groove billed Ani wings and tail" (concept No. 384) and "Groove billed Ani body" (concept No. 1224) in CUB.

Furthermore, as illustrated in the right part of Fig. 4, the final model effectively retains concepts from the generator layer, such as concepts No. 55, No. 412, No. 1216, and No. 1457 in CIFAR100, and concepts No. 384, No. 597, No. 978 and No. 1224 in CUB. Additionally, the final model demonstrates the capability to learn new important concepts. For instance, in CIFAR100, newly

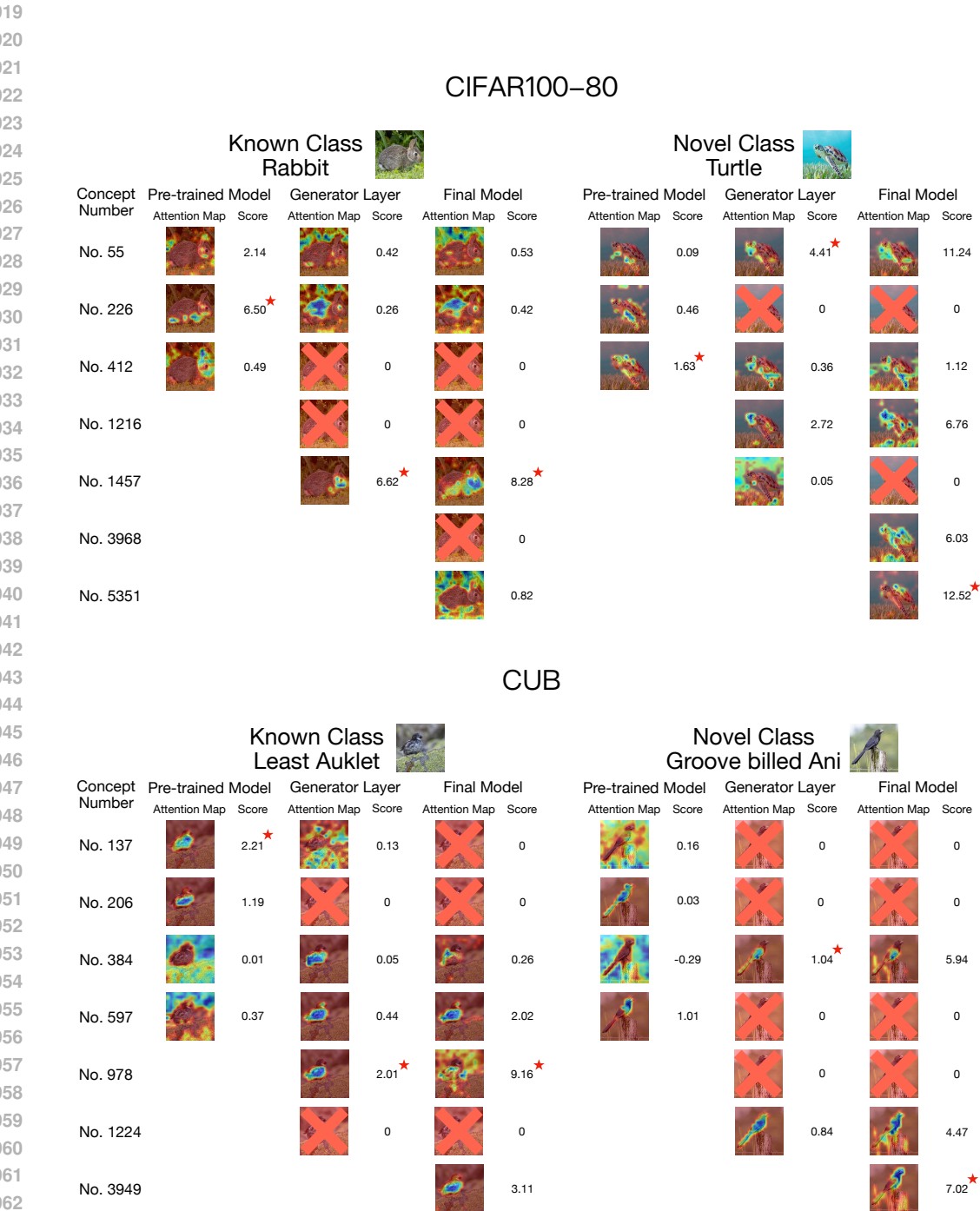

Figure 4: Concept Visualization of Known Class Pre-Trained Model, Generator Layer, and Final Model on CIFAR100-80 and CUB. Attention maps for selected concepts are generated using Grad-CAM(Selvaraju et al., 2017), with model scores provided for each concept. Additionally, ⋆ denotes the highest score among all concepts, while × on the attention map indicates the absence of the model response to that concept. This behavior is exclusive to models utilizing ReLU activation (Generator Layer and Final Model). The blanks in the figure are caused by the fact that the number of concepts learned by the known class pre-trained Model, generator Layer, and final model are different.

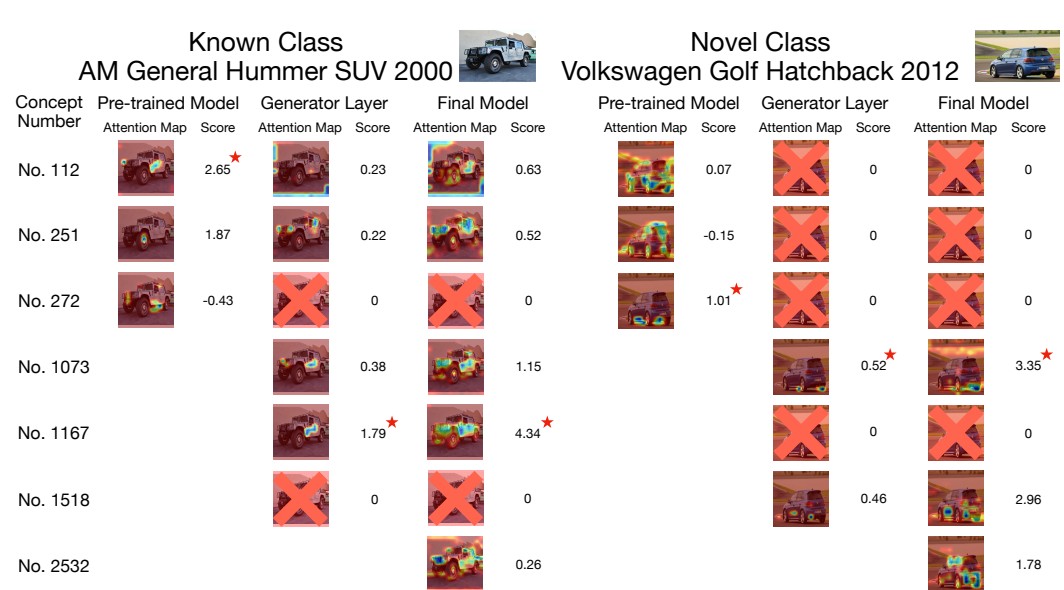

Figure 5: Concept Visualization of Known Class Pre-Trained Model, Generator Layer, and Final Model on Stanford Cars.

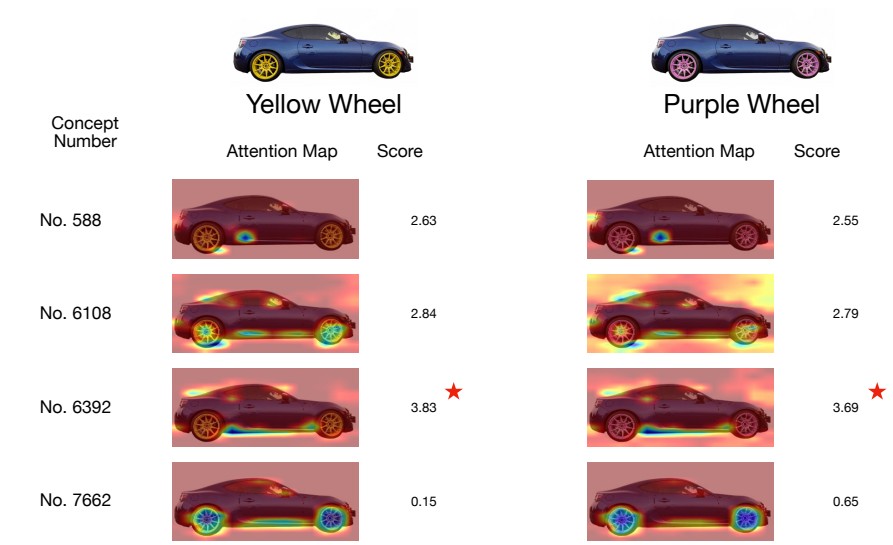

Figure 6: Concept Visualization of the Final Model with Different Colored Wheels on the Same Car.

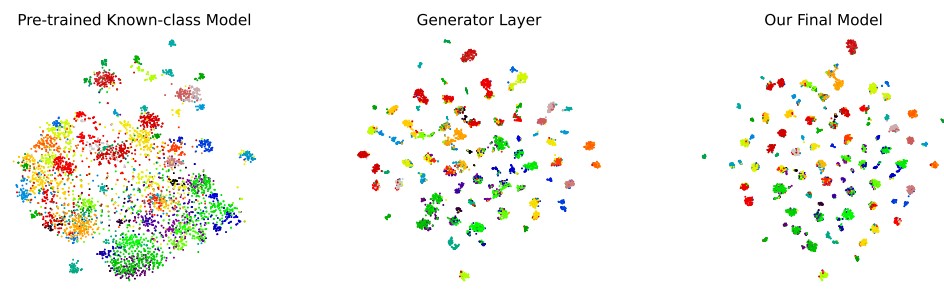

Figure 7: t-SNE visualization on Aircraft.

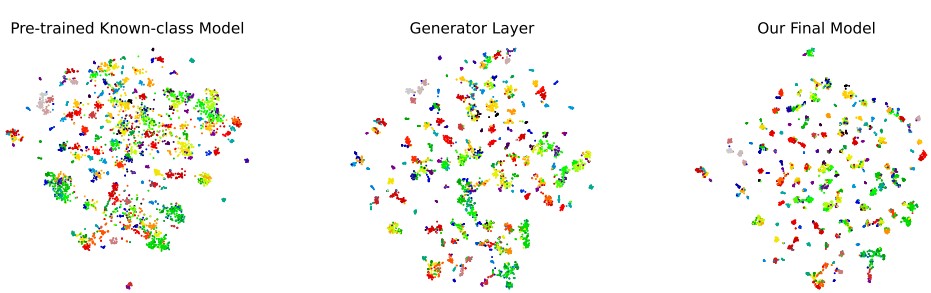

Figure 8: t-SNE visualization on CUB.

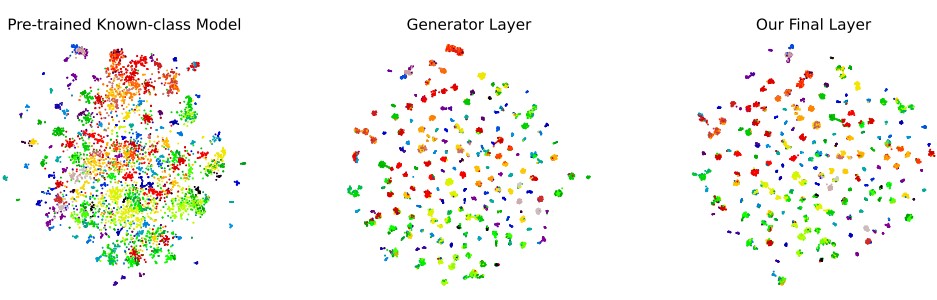

Figure 9: t-SNE visualization on Scars.

learned concept No. 3968 ("head and shell") and concept No. 5351 ("part of shell") exhibit high responses to "turtle" in the final model.

To study the relationship between concepts and visual attributes, we artificially changed the color of the wheels in a car image from yellow to purple and selected several representative concepts for analysis. We observed that the concept scores for non-wheel-related concepts, such as No. 588 and No. 6392, remain largely unchanged. In contrast, some wheel-related concepts exhibit changes, while others do not. For example, concept No. 6108 remains relatively unaffected, while concept No. 7662 shows a noticeable change. This suggests that some wheel-related concepts are sensitive to color changes, while others are not. Additionally, it highlights the robustness of non-wheel-related concepts to alterations in wheel properties.

In summary, these visual results not only affirm our motivation but also validate the importance of each module of our model.

Table 18: The statistical data of the minimal KL divergence between neuron responses in our linear method and those of crNCD and SPTNet. The interval represents the KL Divergence range.

| Method | (0, 0.01) | [0.01, 0.1) | [0.1, 0.2) | [0.2, 0.5) | [0.5, 1.0) | [1.0, ∞) |
|---|---|---|---|---|---|---|
| SPTNet | 13 | 264 | 181 | 190 | 77 | 43 |
| crNCD | 7 | 113 | 141 | 289 | 192 | 26 |

## N MORE TSNE VISUALIZATION

## O ANALYSIS OF NEURONS ACTIVATION

To verify our linear method captures more derivable concepts that SPTNet and crNCD do not, we conduct an additional experiment. Specifically, we analyze the responses of all 768 neurons in the encoder of our linear method, SPTNet, and crNCD using 100 randomly selected samples from the Stanford Cars dataset. The responses of the neurons are converted into probability distributions on these 100 samples using the softmax function. Since concepts are directly linked to neurons, if the concepts learned by two neurons are similar, their probability distributions across the 100 samples should also be similar. To quantify this similarity, we employ the Kullback-Leibler (KL) divergence to compare the probability distributions of our linear method with those of SPTNet and crNCD. For each neuron in our linear method, we calculate the minimum KL divergence with respect to all neurons in SPTNet and crNCD, respectively.

Tab.18 shows the distribution of neurons in our methods based on their minimum KL divergence values against SPTNet and crNCD. Notably, compared to SPTNet, at least 43 neurons in our method are entirely distinct from those in SPTNet (KL divergence > 1.0), and an additional 77 neurons exhibit differences (KL divergence between 0.5 and 1.0). A similar trend is observed when comparing our method with crNCD. These findings demonstrate that our linear method generates some concepts not captured by either SPTNet or crNCD. Thus, we infer that "existing approaches may struggle to capture all the derivable concepts useful for knowledge transfer".

