# OpenReview forum: "Composing Novel Classes: A Concept-Driven Approach to Generalized Category Discovery"
_ICLR.cc/2025/Conference — Submitted to ICLR 2025_

### Official Review · Reviewer_wv58 · 2024-10-17

**Soundness:** 2
**Presentation:** 2
**Contribution:** 2
**Rating:** 5
**Confidence:** 4

**Summary:**

This paper studies the task of Generalized Category Discovery (GCD), which aims to group mixed visual data from base and novel classes by leveraging representations learned from semantically related base classes. This task is closely related to open-world semi-supervised learning. In the preliminary study, the authors observed that existing works might not fully leverage the known knowledge (concepts) from base classes. To this end, this paper proposes a novel framework, ConceptGCD, that uses a covariance loss to learn distinctive concepts to facilitate category discovery. The framework is well-designed to allow the model to use both "derivable concepts" to aid novel class discovery while also enabling the model to learn "underivable concepts" that are customized to novel classes. A covariance loss is used to promote diversity in concept learning. The experiments show consistent improvements over the compared methods on both base and novel classes.

**Strengths:**

- This paper is well-structured.
- The related works are comprehensively discussed.
- Strong performance is achieved across six benchmarks, with the proposed method outperforming previous approaches.
- The comparison of methods is comprehensive.

**Weaknesses:**

**Major:**
- The validity of the experiment is questionable: When DINOv1 (self-supervised on ImageNet) is used as the backbone, the novel classes of CIFAR-100-80 and ImageNet-100-50 are not novel. When DINOv2 (trained on curated ImageNet, CUB, StanfordCars, FGVC) is used as the backbone, except for the novel classes of Herbarium19, all the novel classes in the evaluated datasets are not novel. The backbone was self-supervised on all the classes already (seen). Under this experimental setting, are the "Concepts" for base/novel classes learned via the proposed framework or via the backbone pre-training? I have my doubts about this. One suggestion is that it would be beneficial to apply the proposed method on a ResNet model trained from scratch, as this would isolate the influence of backbone pre-training, helping us better understand the effectiveness of the proposed framework.
- The method requires prior knowledge of the number of clusters. Of course, this is a minor concern, as this limitation is common to most GCD works.
- The novelty of the proposed framework is limited: the major contribution is the application of covariance loss in GCD.

**Minor:**
- The English writing should be improved-i.e., “Despite their progress, our analysis experiments show that novel classes can achieve impressive clustering results on the feature space of a known class pre-trained model, suggesting that existing methods may not fully utilize known class knowledge.” (P1#L14): Novel classes should not be the subject for "achieve impressive clustering results." Instead, the correct logic should be: "Impressive clustering results are achieved on novel classes."
- From Fig. 1 and the introduction section, I could not understand: i) what are "derivable concepts"?; ii) how are they generated? This makes the introduction very vague. It would be better to provide the definition or description about the ``concepts'' because the meaning of "concepts" in this paper's context is different from common semantic concepts.
- The motivational insights claimed in the introduction section are not convincing. To confirm the influence of "derivable concepts," further control experiments and qualitative analysis are needed.
- The notation is unclear: 1) At P4#L193-L194, what do $l$ and $l'$ represent? This is only explained on the next page; 2) In Eq.2, $z_{i}$ should be in bold $\mathbf{z}_{i}$ to represent a vector.

**Questions:**

**Questions:**
- The authors claimed, “Intuitively, these derivable concepts are one of the key reasons why known class data can help the model learn novel class data in GCD problems.” (P1#L50) —> My question is, how is this claim supported or demonstrated? (Such as through previous studies or experiments). NCD/GCD is essentially a clustering problem. The key idea/motivation from DTC [1] is to leverage the representation (prior knowledge) learned from base classes to reduce ambiguity when clustering novel classes. In other words, the “clustering criteria” is defined by the base classes, allowing NCD/GCD methods to group novel data under the criteria defined by the base classes. How do the derivable concepts help in this case, and how do the authors prove it?
- The authors claimed, “This observation suggests that existing approaches may struggle to capture all the derivable concepts useful for knowledge transfer.” (P2#L65) —> Could you please further explain why the results in Table 1 support this claim? Why is the relatively lower performance of SPTNet and crNCD attributed to a "struggle to capture all the derivable concepts"?
- My major question is: How are the "Derivable concepts" defined? From what I understand from the Method section, a "concept" is a scalar value (an element) of an image embedding vector. Why is this defined as a concept?
- Why are the "concepts" denoted as semantic visual cues in Fig. 2? Based on the description in the paper, "a concept" is a scalar value. What makes these concepts represent the semantic visual cues such as "window," "wheel," or "headlight" in Fig. 2?
- Furthermore, since the "concepts" are illustrated in Fig. 2 as visual components (e.g., a wheel), I wonder how the values of "concepts" would change when attributes (e.g., color or shape) change within the same class? To be more specific, for the same car model, the wheel color might change significantly (so-called large intra-class variance challenge in fine-grained visual recognition, e.g., StanfordCars). In this case, how would the "concept" values change?
- Lastly, an open question: The main goal of NCD/GCD is to **discover** novel classes, which are **unknown**. Since novel classes are unknown, how can users/practitioners know the number of novel classes a priori? Could the authors please provide some practical application scenarios that satisfy the GCD setting?

**Some suggestions:**
1. The current paper’s description and definition of "Concepts" are unclear and contain several unverified subjective assumptions, which may confuse readers. The proposed method performs well, so explaining it in a clear, technical manner would make it easier for readers to understand and be convinced. Including too many subjective descriptions without experimental evidence may weaken the credibility of the proposal.
2. The authors could design experiments that do not use DINOv1 or v2 to validate the proposed idea. The absolute performance does not need to be high; the goal is to isolate representations learned from large-scale pretraining and solely validate the effectiveness of the proposed method.
3. More comprehensive motivational experiments should be designed to demonstrate the influence of "concepts."

[1] Kai Han, Andrea Vedaldi, and Andrew Zisserman. Learning to discover novel visual categories via deep transfer clustering. In ICCV, 2019.

---

> ### Author Response · Authors · 2024-11-21
> **Response to reviewer wv58**
>
> Thank you for your detailed and thoughtful feedback. We deeply appreciate the effort you put into reviewing our work, and we value your insights. In the following, we have carefully addressed each of your concerns to the best of our ability.
>
> **W1: Experiments with ResNet Model trained from scratch**
>
> We appreciate the reviewers' suggestions. Following your suggestions, we have conducted experiments using ResNet. We compare our approach with the strong baseline, SimGCD, and present the results as follows:
>
> | Dataset | CIFAR |  |  | ImageNet100 |  |   | CUB200 |     |    |
> | --| -| ---| -| -| --| -| - | -| -|
> | - | All   | Known | Novel | All  | Known | Novel | All | Known | Novel |
> | SimGCD | 52.4  | 63.5  | 30.2  | 32.6 | 75.1 | 11.2 | 14.4  | 21.6 | 10.7 |
> | Ours  | 56.7  | 61.9  | 46.4  | 47.2   | 69.5  | 36.0  | 17.0  | 21.2 | 14.9 |
>
> Our method achieved remarkable improvement in the novel classes on three datasets.  Note that for simGCD, the reported results on ImageNet100 represent the best performance we have achieved so far. It is highly time-consuming to tune for the best hyperparameters.
>
> Furthermore, as you mentioned, fine-grained datasets are considered novel for DinoV1. The superior performance of our method on these datasets also substantiates the effectiveness of our approach.
>
> **W2: prior knowledge of the number of clusters**
>
> Yes, this is a common limitation inherent in GCD. Concurrently, as shown in Table 6, our approach still significantly outperforms previous methods with the estimated number of clusters. Moreover, our method does not show a substantial decrease in performance when compared to scenarios where the number of clusters is known, demonstrating the robustness of our method.
>
> **W3: Contribution**
>
> We believe our major novelty does not lie in the application of covariance loss to GCD. The novelty of our paper can be summarized in the following two points:
>
> 1. Novel Learning Framework. We establish a novel framework based on concepts, analyzing the relationship between novel and known concepts. We classify these concepts into two main categories: derivable and underivable concepts. This analysis leads to our proposal of a novel three-stage modeling framework for learning those concepts.
>
> 2. Two effective techniques. In addition to our model framework, we introduce two methods to enhance the concepts learned by the model. First, we incorporate covariance loss into GCD to improve the learning of concepts. Second, we propose a concept score normalization technique that ensures the model learns derivable and underivable concepts in a more balanced manner.
>
> Those two points are novel in the GCD problem and greatly improve the results.
>
> **W4: English writting**
>
> Thank you for your advice. We have fixed this error in our new version.
>
> **W5: Definition of Concepts**
>
> Please refer to General Response.
>
> **W6 & Q2: motivational insights of derivable concept**
>
> Thank you for your feedback. We believe the motivational insights claimed in the introduction section are convincing. Please first refer to General Response for *Derivable and Underivable Concept* and *Derivable Concepts’ Utility*.
>
> As depicted in Figure 1, in the conventional GCD setting, we train a single linear layer and a cosine classifier on top of a pre-trained model that has been trained on known classes. This classifier is then used to cluster novel classes and classify known classes. Unlike SPTNet and crNCD, which incorporate intricate designs and train more parameters, our linear classifier-based approach delivers performance that is on par with these methods on the CUB and ImageNet100 datasets and surpasses them on the Scars dataset. The exciting results achieved by our simple design of generating derivable concepts demonstrate our motivation.
>
> We appreciate your suggestion and are open to conducting further control experiments and qualitative analyses. Could you please specify the types of experiments and analyses you believe would be most beneficial to validate the influence of 'derivable concepts'?
>
> **W7:  notation is unclear**
>
> $l$ is the number of known class concepts $l = |C^{k}|$. $l’$ is the number of novel class concepts $l’ = |C^u|$. Thanks for your advice. We have changed $z_i$ to $\mathbf{z}_i$
>
> **Q1,Q3: Concerns related to concept**
>
> Please refer to General Response.

---

> ### Author Response · Authors · 2024-11-21
>
> **Q4 & Q5: Concept and Semantic Visual Cues**
>
> The concepts are not equivalent to semantic visual cues. Fig. 2, represents "concepts" as semantic visual cues to aid understanding. The visualization of learned concepts is shown in Appendix M. Additionally, following your suggestion, we change the color of the wheel and illustrate the relationship between concepts and semantic visual cues in Appendix M.
>
> **Q6: Application**
>
> For users, we can adapt the existing method [1] to estimate the number of classes, and then discover novel classes. Once we can estimate the number of classes, we can apply GCD to various settings, especially in scenarios, where exists massive unlabeled data and labeling them is costly. In addition, it has been applied to cell discovery [2].
>
> We sincerely hope the response above addresses your concerns. If not, please feel free to raise any remaining issues, and we will be more than happy to address them.
>
> [1] Sagar Vaze et al, Generalized Category Discovery
>
> [2] Brbic et al. MARS: discovering novel cell types across heterogeneous single-cell experiments

---

> ### Comment · Reviewer_wv58 · 2024-11-21
> **Reviewer wv58 Response to the Authors**
>
> Thank you for providing the clarification and additional experiments.
>
> My concerns are partially addressed in the rebuttal, such as the comparison with non-pretrained backbone (e.g., dinoV1 or dinoV2). The additional experiments added by the Authors show the robustness and good performance of the method proposed in this work.
>
> However, my major concerns remained:
>
> (1) Unclear and misleading definition of "Concepts", "Known class concept", "Derivable concept", and "Underivable concept". This concern is major because it is the foundation of the main motivation of this work. The definition, explaination and justification about "Concept" and "Why concept helps" in the context of GCD is too subjective. If the justification holds, qualitative experiments are needed to support the claim.
>
> *More importantly, the Authors used "We represent "concepts" as semantic visual cues in Fig. 2 just for understanding." in the Global Response to justify the concern related to "Concept and Semantic Visual Cues". I could not agree with this answer because: If the Concepts in this work are not Semantic Visual Cues (e.g., attributes or parts as present in this manuscript), this representation is misleading and does not correspond to the facts.* As a reader, the message I received from Fig. 2 is: "concepts are learned compositional semantic cues. That's why the concepts learned from known might be useful for novel classes". However, this is not in line with the actual concept definition in this work.
>
> (2) My question `The authors claimed, “This observation suggests that existing approaches may struggle to capture all the derivable concepts useful for knowledge transfer.” (P2#L65) —> Could you please further explain why the results in Table 1 support this claim? Why is the relatively lower performance of SPTNet and crNCD attributed to a "struggle to capture all the derivable concepts"?`  is not answered. From what observation or experiments the Authors can get this conclusion? In addition, the Authors described " Unlike SPTNet and crNCD, which incorporate intricate designs". I do not think SPTNet and crNCD have intricate designs. And, this is not related to my qustion.
>
> (3) The justification about Novelty. Such two stage learning framework has been widely used with different loss function and architectural designs from the very beginning of GCD and NCD literature because the Base/Novel split setting [1, 2]. Even the foundation NCD work AutoNovel (RankStats) [1] used such learning framework.  Therefore, this framework used in this work can not be considered as a novelty. Thus, the novelty of this work is limited to the loss function.
>
> In conclusion, the idea for tackling GCD presented in this work is intriguing and holds potential for further exploration. I acknowledge it. However, the current manuscript requires substantial improvements in clarity, soundness, and novelty. Additionally, the claims made in this study need to be more thoroughly substantiated, as they are not sufficiently convincing in the present revision.
>
> Therefore, I decided to keep my initial rating. Thank you for the Authors' responses and effort made during the rebuttal stage, which I believe will make this manuscript strongger in its future revision.
>
>
> [1] Han, K., Rebuffi, S. A., Ehrhardt, S., Vedaldi, A., & Zisserman, A. (2021). Autonovel: Automatically discovering and learning novel visual categories. IEEE Transactions on Pattern Analysis and Machine Intelligence, 44(10), 6767-6781.
>
> [2] Wang, H., Vaze, S., & Han, K. (2024). SPTNet: An efficient alternative framework for generalized category discovery with spatial prompt tuning. arXiv preprint arXiv:2403.13684.

---

> ### Author Response · Authors · 2024-11-23
>
> We thank the reviewer for prompt response and would like to further address the remain concerns
>
> **Q1: concepts**:
>
> 1. We employ the term "concept" to denote the semantic patterns learned from data at a subcategory level. Some of these patterns are semantic visual cues that are comprehensible to humans (such as parts & attributes), while others may not be easily interpretable but are stable visual patterns. For instance, prior works [1, 2, 3] have shown that inputs maximizing a neuron’s activation include both recognizable visual semantic cues and abstract visual patterns. In the comments, we clarify that 'the concepts are not equivalent to semantic visual cues', which means our concepts include but not limit to semantic visual cues. We apologize for any confusion that may have arisen.
> 2. We want to emphasize that the effectiveness of our framework is built upon the property of those learned subcategorical patterns instead of whether we denote them as 'concepts'. Our framework operates on the premise that known and novel classes share some common subcategorical semantic patterns, an assumption widely accepted in the field of NCD and GCD, and that novel classes possess some unique semantic patterns. We adopt this terminology by following [1, 6] and agree that this term may have varying meanings under different contexts. To avoid confusion, we can easily replace 'concepts' with other terms such as 'semantic patterns', as it refers to a stable pattern learned from data.
> 3. Our Figure 2 is primarily for illustrative purposes.
>
>
> [1] Nguyen A, Dosovitskiy A, Yosinski J, et al. Synthesizing the preferred inputs for neurons in neural networks via deep generator networks[J]. Advances in neural information processing systems.
>
> [2] Erhan D, Bengio Y, Courville A, et al. Visualizing higher-layer features of a deep network[J].
>
> [3] Matthew D Zeiler and Rob Fergus. Visualizing and understanding convolutional networks.
>
>
> **Q2: struggle to capture all the derivable concepts**
>
> Given the semantic patterns learned from known classes, we define 'derivable concepts' as a linear combinations of those patterns, which can be modeled as a linear layer. We refer to those patterns of the novel classes that cannot be represented as such linear combinations as 'underivable concepts'.
>
> In Fig. 1 and Tab. 1, we use a simple linear layer to combine known class concept scores to generate new concept scores. Based on our definition of derivable concepts, we can generate them by adding a linear layer after the encoder of a known class pre-trained model. We observe that this linear model outperforms SPTNet and crNCD on certain datasets. We believe this performance improvement arises because our linear method captures concepts that SPTNet and crNCD do not.
>
> To verify this, we conduct an additional experiment. Specifically, we analyze the responses of all 768 neurons in the encoder of our linear method, SPTNet, and crNCD using 100 randomly selected samples from the Stanford Cars dataset. The responses of the neurons are converted into probability distributions on these 100 samples using the softmax function. Since concepts are directly linked to neurons, if the concepts learned by two neurons are similar, their probability distributions across the 100 samples should also be similar. To quantify this similarity, we employ the Kullback-Leibler (KL) divergence to compare the probability distributions of our linear method with those of SPTNet and crNCD. For each neuron in our linear method, we calculate the minimum KL divergence with respect to all neurons in SPTNet and crNCD, respectively.
>
> The table below shows the distribution of neurons in our methods based on their minimum KL divergence values against SPTNet and crNCD. Notably, compared to SPTNet, at least 43 neurons in our method are entirely distinct from those in SPTNet (KL divergence > 1.0), and an additional 77 neurons exhibit differences (KL divergence between 0.5 and 1.0). A similar trend is observed when comparing our method with crNCD. These findings demonstrate that our linear method generates some concepts not captured by either SPTNet or crNCD. Thus, we infer that “existing approaches may struggle to capture all the derivable concepts useful for knowledge transfer”. We hope this additional experiment addresses your concern.
>
> | Method  | (0, 0.01) | [0.01, 0.1) | [0.1, 0.2) | [0.2, 0.5) | [0.5, 1.0) | [1.0, ∞) |
> |---------|-----------|------------|------------|------------|------------|----------|
> | SPTNet  | 13        | 264        | 181        | 190        | 77         | 43       |
> | crNCD   | 7         | 113        | 141        | 289        | 192        | 26       |
>
> Finally, we would like to apologize for the irrelevant statement: "Unlike SPTNet and crNCD, which incorporate intricate designs." Thank you for pointing this out.

---

> ### Author Response · Authors · 2024-11-23
>
> **Q3: Novelty**
>
> Our framework consists of three distinct phases, each designed to efficiently learn derivable and underivable concepts. Here’s a detailed description of each phase:
> 1. **Representation Initialization**: We begin by initializing representations for known classes, similar to existing methods [1,2]. This sets the foundation for further learning.
> 2. **Learning Derivable Representations**: This phase introduces a novel approach not found in existing work[1,2]. We employ a generator (linear) layer specifically to generate derivable concepts. The effectiveness of this approach is demonstrated in Table 1, highlighting its significance in our framework.
> 3. **Simultaneously Learning Derivable and Underivable Concepts**: In this final stage, we fine-tune both the representation and the classifier. Unlike previous methods [1,2] that directly tune representations, which are ineffective in learning derivable concepts (as shown in Fig 1 and Tab 1), we have designed a contrastive learning strategy to preserve the derivable concepts learned in the second phase. This strategy is crucial, as it prevents the model from losing these learned concepts. Additionally, we adopt an expansion (linear) layer to broaden the concept space and propose a concept normalization technique to better learn underivable concepts.
>
> This structured approach ensures that our framework effectively captures and utilizes both derivable and underivable concepts, enhancing overall performance.
>
> [1] Han, K., Rebuffi, S. A., Ehrhardt, S., Vedaldi, A., & Zisserman, A. (2021). Autonovel: Automatically discovering and learning novel visual categories. IEEE Transactions on Pattern Analysis and Machine Intelligence, 44(10), 6767-6781.
>
> [2] Wang, H., Vaze, S., & Han, K. (2024). SPTNet: An efficient alternative framework for generalized category discovery with spatial prompt tuning. arXiv preprint arXiv:2403.13684.

---

> ### Comment · Reviewer_wv58 · 2024-11-23
> **Reviewer wv58 Response to the Authors**
>
> Thanks the Authors for providing further clarification on the definition of "concepts" and novelty.
>
> I would like to clarify that, I understood well what does "concepts" represent in this work, from a technical perspective. And I acknowledge its effectiveness. My major concern is that, to clarify this, the current manuscript indeed needs major revision, especially the Introduction part. This should be motivated from technical perspective, with experimental observation as support (like you did in the paper and with the additional experiments).
>
> The current writing is too subjective. As you mentioned, "Some of these patterns are semantic visual cues that are comprehensible to humans (such as parts & attributes), while others may not be easily interpretable but are stable visual patterns. ", if this is the case, showing a visualization in the Introduction to motivate and explain your motivation from technical perspective would be very clear. However, this really needs major revision.
>
> **I stress again, I like the idea of using these fine-grained relevant and irrelevant middle-level semantics to improve GCD/NCD. This is novel.**
>
> However, the major novelty is limited to the losses. This is the fact that can be hardly updated during the rebuttal period. As you listed in your response, NCD/GCD usually starts with supervised representation (learned on known or dino initialized). So Step 1 is common practice in NCD/GCD work by default, because this is the core idea of NCD/GCD. This should be included in your contribution. Further, Step 2 and Step 3 are widely used paradigm with different loss function designs or knowledge distillation schemes, since essentially, NCD/GCD is a transfer learning task. Thus, from framework design perspective, I really cannot consider Step 2 and 3 as your contribution.
>
> The major novelty in my understanding is: 1) the idea of leverage derivable and derivable concepts to improve GCD; 2) the proposal of the loss functions to achieve this intuitive idea.  I consider it is limited still. Thus, I keep my initial score.
>
> Thank you!

---

> ### Author Response · Authors · 2024-11-25
>
> **Writting**
>
> Thank you for your thoughtful feedback. We have added additional visualizations of concepts in the appendix. Additionally, we would like to clarify the logical flow of our Introduction to demonstrate that our paper is structured, objective, and aligned with its primary focus. Our reasoning is as follows:
> 1. **The model can learn concepts**:
> This is well-supported by extensive literature [1, 2, 3, 4, 5, 6]. While we agree that visualizations and in-depth analyses are valuable, they are not the primary focus of this work. A detailed exploration of concepts is more appropriate for studies dedicated to visualization and interpretability. Our primary objective is to address the Generalized Category Discovery (GCD) problem effectively. For this reason, we believe including concept visualizations as technical evidence in the appendix is more suitable than placing them in the main text.
> 2. **Similar semantic patterns of novel and known class concepts**:
> Our assumption—common in many GCD studies—is that some novel class concepts share similar semantic patterns with known class concepts. This foundational assumption does not require additional substantiation. Building on this, we categorized concepts into two types: derivable and underivable concepts.
> 3. **The necessity of derivable concepts**:
> The necessity of derivable concepts: As demonstrated in Table 1, we highlight that a model focusing on capturing derivable concepts can achieve performance comparable to or even surpass current SOTA methods [7,8] on certain datasets. To investigate the underlying reasons behind this performance, we conducted an additional experiment based on your suggestion. The results reveal that some neurons in our model exhibit distinctly different activation patterns compared to those in existing SOTA methods, indicating that our approach captures unique concepts absent in prior methods. This observation, combined with the evidence from Table 1, leads us to conclude that current SOTA methods fail to fully capture derivable concepts. Furthermore, the results also demonstrate that these concepts are crucial for enhancing model performance. We would also like to apologize for the subjective phrasing in our original manuscript, particularly the statement: “This observation suggests that existing approaches may struggle to capture all the derivable concepts useful for knowledge transfer.” Based on your feedback, we have refined this section in the updated version of our manuscript. We hope the revised text addresses your concerns and provides a clearer justification for our approach.
> 4. **Our model proposal**:
> Building on the above assumption and findings of this experiment, we propose our model, which is specifically designed to better learn both derivable and underivable concepts.
> We hope this clarification addresses your concerns and provides a clear overview of our paper’s logical flow. Thank you again for your valuable insights, which have been instrumental in improving our work.
>
> [1] Lee J H, et al. From neural activations to concepts: A survey on explaining concepts in neural networks.
>
> [2] Erhan D, et al. Visualizing higher-layer features of a deep network.
>
> [3] Nguyen A, et al. Understanding neural networks via feature visualization: A survey[J]. Explainable AI: interpreting, explaining and visualizing deep learning.
>
> [4] Nguyen A, et al. Synthesizing the preferred inputs for neurons in neural networks via deep generator networks
>
> [5] D. Bau et al, Network Dissection: Quantifying Interpretability of Deep Visual Representations
>
> [6] Matthew D Zeiler et al. Visualizing and understanding convolutional networks.
>
> [7] Peiyan Gu, Chuyu Zhang, Ruijie Xu, and Xuming He. Class-relation knowledge distillation for novel class discovery.
>
> [8] Hongjun Wang, Sagar Vaze, and Kai Han. Sptnet: An efficient alternative framework for generalized category discovery with spatial prompt tuning.

---

> ### Author Response · Authors · 2024-11-25
>
> **Novelty**
>
> As you mentioned, “Step 2 and Step 3 are widely used paradigms with different loss function designs or knowledge distillation schemes, since essentially, NCD/GCD is a transfer learning task.” We respectfully disagree with this characterization. To the best of our knowledge, our framework is the first in GCD to adopt a multi-stage methodology that explicitly separates the learning of derivable and underivable concepts.
> In NCD, methods such as AutoNovel [1] and crNCD [3] typically follow a two-stage process: supervised pretraining on known classes, followed by joint training on all classes. In contrast, recent GCD approaches, such as GCD [5], SPTNet [2], and SimGCD [4], generally employ a single-stage learning model, bypassing supervised pretraining entirely. These approaches attempt to learn derivable and underivable concepts simultaneously, which, as shown in Table 1, is less effective.
> In detail, our derivable-underivable separate learning methodology distinguishes itself from existing works in three key aspects: Purpose, Framework, and Loss Function.
> 1. Purpose:
>   - Step 2: The primary objective is to focus exclusively on learning derivable concepts, a goal that has not been explored in any existing GCD research [1, 2, 3, 4, 5].
>   - Step 3: The objective is to retain the derivable concepts learned in Step 2 while simultaneously learning new underivable concepts. This purpose stems directly from our strategy of separating the learning processes for derivable and underivable concepts, making it unique in the context of GCD research[1, 2, 3, 4, 5].
> 2. Framework:
>   - Step 2: We introduce a novel linear layer (generator layer) to explicitly model the relationship between derivable and known class concepts. This layer design is distinct from existing GCD approaches[1, 2, 3, 4, 5].
>   - Step 3: The framework expands the linear layer to effectively incorporate underivable concepts. Model growth is not a common practice in existing GCD methods [1, 2, 3, 4, 5].
> 3. Loss Function:
>   - Step 2: Our loss function incorporates a novel covariance loss, which you acknowledge as a significant contribution.
>   - Step 3: We utilize contrastive knowledge distillation loss to retain derivable concepts and introduce a concept score normalization mechanism to balance the learning of derivable and underivable concepts—addressing a unique challenge in this phase. The overall loss function is specifically designed for Step 3 and is distinct from any current GCD work [1, 2, 3, 4, 5].
>
> These innovations collectively establish the novelty of our Step 2 and Step 3 designs. If you still believe that Steps 2 and 3 are widely used, we would greatly appreciate it if you could point us to specific studies or works that support this claim.
>
> [1] Han, K., Rebuffi, S. A., Ehrhardt, S., Vedaldi, A., & Zisserman, A. (2021). Autonovel: Automatically discovering and learning novel visual categories. IEEE Transactions on Pattern Analysis and Machine Intelligence, 44(10), 6767-6781.
>
> [2] Wang, H., Vaze, S., & Han, K. (2024). SPTNet: An efficient alternative framework for generalized category discovery with spatial prompt tuning. arXiv preprint arXiv:2403.13684.
>
> [3] Gu et al. Class-relation Knowledge Distillation for Novel Class Discovery, ICCV 2023.
>
> [4] Wen et al. Parametric classification for generalized category discovery: A baseline study, ICCV2023.
>
> [5] Vaze et al. Generalized category discovery, CVPR2022.

---

> ### Comment · Reviewer_wv58 · 2024-11-27
> **Reviewer wv58 Response to the Authors**
>
> Dear Authors,
>
> Thanks for thr further clarification.
>
> As a reader, I still find the writing is subjective without experimental support, especially for the definition of "concepts" in this work. This is also pointed out by Reviewer Au3V.
>
> Regarding the novelty, the proposed losses decide the shape (stages) of the current framework. So, I cannot consider the framework as an individual novelty. Further, I did not say prior work used exact the same framework as this work. It is similar. One more stage designed for the corresponding loss does not make the framework per se novel.
>
> Thank you!
>
> Reviewer wv58

---

### Official Review · Reviewer_Au3V · 2024-10-27

**Soundness:** 2
**Presentation:** 3
**Contribution:** 3
**Rating:** 5
**Confidence:** 4

**Summary:**

This paper addresses the GCD problem by introducing ConceptGCD. The paper mentioned that this framework can enhance class discovery by categorizing concepts as either derivable or underivable from known classes and learning them in stages by using a generator layer with a covariance-augmented loss and a concept score normalization strategy

**Strengths:**

1. The paper is well-organized and clearly written, making the methodology straightforward to follow.

2. It presents a novel approach, ConceptGCD, which introduces a concept categorization strategy to assist with novel class discovery in generalized category discovery.

**Weaknesses:**

1. The abstract mentions that "analysis experiments show that novel classes can achieve impressive clustering results on the feature space of a known class pre-trained model," Is the corresponding evidence from Table 1? It's unclear to me, would you give more explanation on it?

2. There is no clear definition or description of "concept"（mentioned in the introduction and method）  and its distinction from class in the paper, nor are "derivable" and "underivable" concepts (mentioned in both introduction and method) adequately defined. I would suggestion to provide detailed definitions and examples of derivable and underivable concepts.

3. The definitions of key variables \( n \) and \( m \) in lines \#289-290 are unclear, yet they appear important to the framework. Additionally, there is no ablation study on the impact of \( n \) and \( m \).  Please provide clear definitions of n and m, explain their significance to the framework, and conduct an ablation study to demonstrate their impact on the model's performance.

4. The generator and expansion layers increase the model size, and it is unclear if the performance gains are due to the larger model rather than the proposed method. The baseline models need to be scaled similarly to ensure fair comparisons. I suggest conducting additional experiments with scaled baseline models to isolate the effect of their proposed method from the impact of increased model size.

**Questions:**

1. Clarification on Line #41-31: The paper states that “sharing encoder makes it challenging to transfer knowledge between labeled and unlabeled data and can be susceptible to the label uncertainty of unlabeled data.” However, crNCD focus on transferring class-relation information between known and novel classes, without suggesting that shared encoders are susceptible to label uncertainty in unlabeled data. Could you clarify this interpretation?

2. Proof of Derivable Concepts’ Utility: The paper claims, “Intuitively, these derivable concepts are one of the key reasons why known class data can help the model learn novel class data in GCD problems.” Could you provide experimental or theoretical proof for this statement? Additionally, the concept is similar to existing works [1,2]; a comparison would strengthen the paper.

[1] Meta Discovery: Learning to Discover Novel Classes Given Very Limited Data
[2] Supervised Knowledge May Hurt Novel Class Discovery Performance

3. Details on the Linear Layer in the Derivable Concept Generator: The paper describes a linear layer and a ReLU layer after the encoder for derivable concept generation, trained with a covariance-augmented loss. How large is this linear layer? Given that larger models typically improve performance, could this impact baseline comparisons?

---

> ### Author Response · Authors · 2024-11-21
> **Response to Reviewer Au3V**
>
> Thank you for your thoughtful feedback. We sincerely appreciate your effort and have addressed your questions below.
>
> **W1: More explanation**
>
> As depicted in Figure 1, following the conventional GCD setting [1,2], we train a single linear layer and a cosine classifier on top of a pre-trained model that has been trained on known class data. Unlike SPTNet and crNCD, which incorporate intricate designs and train more parameters, our simple approach delivers performance that is on par with these methods on the CUB and surpasses them on ImageNet100 datasets and the Scars dataset in Table 1.
>
> [1] Sagar Vaze, Kai Han, Andrea Vedaldi, and Andrew Zisserman. Generalized category discovery.
> [2] Xin Wen, Bingchen Zhao, and Xiaojuan Qi. Parametric classification for generalized category discovery: A baseline study.
>
> **W2 & Q2: definition or description of "concept"**
>
> Please refer to General Response.
>
> **W3:  definitions of key variables m and n**
>
> m is the number of concepts in the second stage model and n is the number of all concepts in our final model. Due to the model design and our ``concept’’ definition, m is also the output dimension of the generator layer and n is also the output dimension of the expansion layer. The ablation study of n and m are provided in Appendix G and H.
>
> **W4 & Q3: Mode Size**
>
> The generator and expansion layers increase the model size, and it is unclear if the performance gains are due to the larger model rather than the proposed method. The baseline models need to be scaled similarly to ensure fair comparisons. I suggest conducting additional experiments with scaled baseline models to isolate the effect of their proposed method from the impact of increased model size.
>
> | Model       | CIFAR100 |                  | CUB    |                  | StanfordCars |                  | Aircraft |                  |
> | ----------- | -------- | ---------------- | ------ | ---------------- | ------------ | ---------------- | -------- | ---------------- |
> |             | #Param   | #Trainable Param | #Param | #Trainable Param | #Param       | #Trainable Param | #Param   | #Trainable Param |
> | simGCD      | 92.2M    | 13.5M            | 92.3M  | 13.5M            | 92.3M        | 13.5M            | 92.2M    | 13.5M            |
> | infoSieve   | 85.8M    | 7.1M             | 85.8M  | 14.2M            | 85.8M        | 14.2M            | 85.8M    | 7.1M             |
> | SPTNet      | 92.4M    | 13.7M            | 92.5M  | 13.8M            | 92.5M        | 13.8M            | 92.4M    | 13.7M            |
> | Ours (n=2m) | 88.5M    | 9.8M             | 88.8M  | 10.1M            | 88.8M        | 10.1M            | 88.5M    | 9.8M             |
> | Ours (n=5m) | 92.5M    | 13.8M            | 93.2M  | 14.5M            | 93.2M        | 14.5M            | 92.5M    | 13.8M            |
>
> The parameter counts for the models are illustrated above. At n=5m, our model's parameter count is nearly on par with those of SimGCD and SPTNet. However, at n=2m, our model boasts a substantially smaller parameter count compared to the others, except InfoSieve, which adopts k-means to cluster all classes. In our final model (n=5m, m=1536), the parameters of the linear and classifier layers are calculated as 768×7680+7680×number of classes. On the other hand, SimGCD and SPTNet employ a DINO head, which is composed of four MLP layers.
>
> | Model                   | CIFAR100 |          |          | CUB      |          |          | Stanfordcars |          |          | Aircraft |          |          |
> | -- | --- | ----- | --| --| --| ----| ----| --- | --| ----| --| --|
> | | All | Known | Novel | All | Known | Novel |All | Known | Novel |All | Known | Novel |
> | SimGCD                  | 78.1     | 77.6     | 78.0     | 60.3     | 65.6     | 57.7     | 46.8         | 64.9     | 38.0     | 48.8     | 51.0     | 47.8     |
> | InfoSieve               | 78.3     | 82.2     | 70.5     | 69.4     | 77.9     | 65.2     | 55.7         | 74.8     | 46.4     | 56.3     | 63.7     | 52.5     |
> | SPTNet                  | 81.3     | 84.3     | 75.6     | 65.8     | 68.8     | 65.1     | 59.0         | 79.2     | 49.3     | 59.3     | 61.8     | 58.1     |
> | ConceptGCD (Ours，n=2m) | 82.7     | 84.2     | 79.6     | 68.8     | 72.9     | 66.8     | 70.2         | 81.0     | 65.0     | 59.9     | 57.7     | 61.1     |
> | ConceptGCD (Ours，n=5m) | **82.8** | **84.1** | **80.1** | **69.4** | **75.4** | **66.5** | **70.1**     | **81.6** | **64.6** | **60.5** | **59.2** | **61.1** |
>
> The results indicate that when n=2m, our method also outperforms previous approaches, except on the CUB dataset, where we are slightly inferior to InfoSieve. These findings suggest that our improvements are not mainly due to an increase in parameter count, but rather the inherent effectiveness of our method.
>
> We sincerely hope the response above addresses your concerns. If not, please feel free to raise any remaining issues, and we will be more than happy to address them.

---

> ### Author Response · Authors · 2024-11-21
>
> **Q1 Clarification on Line #41-31**
>
> In crNCD, it has been observed that the existing strategy of sharing encoders can disrupt the existing class relations, impeding the effective transfer of knowledge between labeled and unlabeled data. We hypothesize that the reason for this disruption can be susceptible to the label uncertainty of unlabeled data. Therefore, we assert that 'sharing encoders make it challenging to transfer knowledge between labeled and unlabeled data and can be susceptible to the label uncertainty of unlabeled data.'
> To reduce misunderstanding, we will revise the original statement and remove our hypothesis: 'However, as demonstrated in crNCD (Gu et al., 2023), the strategy of sharing encoders undermines meaningful class relations, complicating the transfer of knowledge between labeled and unlabeled data.'

---

> > ### Author Response · Authors · 2024-11-25
> > **Reminder for feedback**
> >
> > Thank you once again for the reviewer’s constructive suggestions. As the discussion section will be concluded soon, we sincerely ask if our responses have addressed your concerns. If so, would you consider raising your score? If not, please feel free to raise any further questions; we would be glad to address your concerns in more detail.

---

### Official Review · Reviewer_Me2K · 2024-11-03

**Soundness:** 3
**Presentation:** 2
**Contribution:** 2
**Rating:** 5
**Confidence:** 5

**Summary:**

This paper studies the problem of Generalized Category Discovery, which aims to assign unlabeled samples with both known and unknown class clusters.
To tackle this problem, they propose a concept-based framework, ConceptGCD, which introduces a concept generation module and a novel covariance loss to derive new concepts of novel classes from known classes, and consequently learn independent underivable concepts for novel classes. They evaluate the proposed ConceptGCD on six GCD benchmarks and demonstrate the effectiveness of their method.

**Strengths:**

S1. They design a novel two-branch concept-based architecture, which introduces a novel shared derivable concept learning branch that consists of a frozen labeled known classes pre-trained encoder to capture concepts of known classes, which are then used to generate the derivable concept of novel classes. With this shared branch, the model can boost the representation learning of unlabeled samples with concepts learned from known classes.

S2. For the final model to be fine-tuned jointly, they expand the derivable concept generate layer to learn more independent concepts of novel classes with proposed knowledge transfer constraint, and a normalization term is introduced to balance the learning process between labeled and unlabeled samples.

S3. They elaborately design a three-stage training strategy for the proposed ConceptGCD to alleviate the influence of the noise introduced by the uncertainty of unlabeled samples.

S4. They provide sufficient analysis of each component of proposed ConceptGCD, includes detailed ablation study on each loss and neuron layer, and visualization of distribution of whole representation space and concept.

**Weaknesses:**

W1. As mentioned in Line 243, “When the batch size B is sufficiently large, ...”, could you show the performance tendency when batch size is changed, especially when it is small?

W2. The proposed covariance loss is only implied on top-m dimensions of the feature, and there is no explicit constraint for other n-m dimensions of the feature, why do these underivable concepts have no constraint and how does this loss influence these parts of the feature?

W3. The implement detail of the set of negative samples in Line 285, is not expressed in this paper.

W4. Could you try to explain the significant gains achieved in Stanford Cars? and why this dataset is not shown in the T-SNE and the concept visualization?

W5. In Tab. 12, when EL dim = 10.0m the performance remains comparable to EL dim = 5.0m, except for CUB, could you please analyze this phenomenon?

W6. Could you do the mean-variance analysis for the main results to show the stability of the proposed method?

**Questions:**

Please justify the issues in Weaknesses.

---

> ### Author Response · Authors · 2024-11-21
> **Response to Me2K: Part A**
>
> Thank you for your detailed feedback, which is crucial for enhancing the quality of the manuscript. Below, we address the concerns raised and hope to resolve them effectively.
>
> **W1: Batch Size**
>
> Thank you for your feedback. The experiment results regarding the hyperparameter $B$ are presented in the table below. As illustrated, the model performance deteriorates when the batch size is too small. There are two potential reasons for this. First, a small batch size may not provide a reliable approximation of the covariance between concepts $i$ and $j$ for $C_{i,j}$. Second, a reduced batch size can adversely affect the regularization constraint in the unsupervised loss term $\mathcal{L}_{u}$ (refer to Eq. 9 in the Appendix), leading to poor clustering performance. Furthermore, the table indicates that a larger batch size benefits the model's performance. In this study, we adhere to the settings of previous works [1, 2] and select a batch size of 128.
>
> | Batch size | CIFAR100 |  |  | CUB  |   |   | StanfordCars | | | Aircraft | | |
> | ---------- | -------- | -------- | -------- | ---- | ---- | ---- | ------------ | ------------ | ------------ | -------- | -------- | -------- |
> | - | All | Known | Novel |  All | Known | Novel | All | Known | Novel |  All | Known | Novel |
> | 32         | 73.7     | 79.8     | 61.5     | 54.7 | 57.3 | 53.4 | 64.7         | 76.2         | 59.2         | 56.4     | 60.4     | 54.4     |
> | 64         | 75.6     | 82.0     | 63.0     | 64.5 | 74.1 | 59.7 | 68.6         | 80.1         | 63.0         | 58.7     | 62.3     | 56.9     |
> | 128        | 82.8     | 84.1     | 80.1     | 69.4 | 75.4 | 66.5 | 70.1         | 81.6         | 64.6         | 60.5     | 59.2     | 61.1     |
> | 256        | 83.2     | 83.8     | 81.9     | 69.7 | 70.4 | 69.3 | 70.0         | 81.5         | 64.5         | 60.8     | 59.6     | 61.4     |
> | 512        | 83.3     | 83.7     | 82.6     | 69.7 | 71.0 | 69.1 | 70.0         | 82.7         | 63.9         | 61.0     | 59.4     | 61.8     |
>
> [1] Sagar Vaze, Kai Han, Andrea Vedaldi, and Andrew Zisserman. Generalized category discovery.
>
> [2] Xin Wen, Bingchen Zhao, and Xiaojuan Qi. Parametric classification for generalized category discovery: A baseline study.
>
> **W2: covariance loss**
>
> The covariance loss can be applied across all dimensions, not just limited to the top-m dimensions. As evidenced in Table 5, during the second stage of our experiments, we incorporate the covariance loss across all dimensions (dim=m) and discover its beneficial effects. However, upon reviewing the results in Table 8, we observe that including covariance loss in the third stage (dim=n) yielded only marginal improvements. Consequently, we have elected to exclude this loss in the third stage of our approach. We have adjusted the relevant description to avoid misunderstanding.
>
> **W3: implement detail of the set of negative samples**
>
> As in traditional contrastive learning [1], we treat all other instances as negative samples without any hard example mining strategy. In detail, the memory buffer contains 2048 negative samples.
>
> [1] He et al. Momentum Contrast for Unsupervised Visual Representation Learning.
>
> **W4: Explain of Stanford Cars**
>
> As shown in Tab. 5 of the paper, the model's impressive performance on Stanford Cars can be primarily attributed to the effectiveness of $\mathcal{L}_{cov}$. This loss function proves beneficial across nearly all datasets, and it is particularly impactful for Stanford Cars. This effectiveness may be linked to the specific characteristics of the dataset itself.
>
> We have included the visualization of Stanford Cars (t-SNE and concept) in Appendix M and N.
>
> **W5. CUB performance w.r.t EL dimension**
>
> An overly large expansion layer can cause the model to focus excessively on learning underivable concepts, which are difficult to learn and prone to devolving into noise. As the dimension $m$ of the expansion layer increases, the model's propensity to capture noise also increases, which can disrupt the model's existing knowledge and degrade its performance. While concept score normalization can mitigate this issue to some extent, it is not sufficient to completely offset the impact of a significantly large expansion layer. As a result, the model may still learn a considerable amount of noise, leading to a noticeable decline in performance.
>
> To test this hypothesis, we conduct an additional experiment with an even larger expansion layer, specifically one with 20m dimensions. As the results in Table 12 demonstrate, the model's performance deteriorates across nearly all datasets.

---

> > ### Author Response · Authors · 2024-11-21
> > **Response to Me2K: Part B**
> >
> > **W6. Mean-variance analysis**
> >
> > We conduct all of our experiments three times, except for the ImageNet dataset due to the high computational and time costs. The results present in the article are averages from these three experiments. Variances are provided in the following table and have been updated in Appendix K to demonstrate the stability of the proposed method.
> >
> > | Dataset      | All  | Seen | Novel |
> > | ------------ | ---- | ---- | ----- |
> > | CIFAR100     | 0.09 | 0.12 | 0.45  |
> > | CUB          | 0.61 | 1.17 | 0.33  |
> > | StanfordCars | 0.52 | 1.18 | 1.25  |
> > | Aircraft     | 1.30 | 0.36 | 2.13  |
> >
> >
> > We sincerely hope the response above addresses your concerns. If not, please feel free to raise any remaining issues, and we will be more than happy to address them.

---

> > > ### Author Response · Authors · 2024-11-25
> > > **Reminder for feedback**
> > >
> > > Thank you once again for the reviewer’s constructive suggestions. As the discussion section will be concluded soon, we sincerely ask if our responses have addressed your concerns. If so, would you consider raising your score? If not, please feel free to raise any further questions; we would be glad to address your concerns in more detail.

---

> > > > ### Comment · Reviewer_Me2K · 2024-11-28
> > > > **Acknowledgement and Concerns about the novelty**
> > > >
> > > > Thanks for the authors' efforts. The responses partially address my concerns. Although the proposed concept-driven method is effective for GCD, the step-wise learning approach is not new. It has been explored in other areas, especially for the extension of the FC layer, like incremental/continual learning [1]. After reading the comments from other reviewers, I agree that the dimensional extension is not equal to concept extension. After overall consideration, I keep my initial rate.
> > > >
> > > >
> > > > [1] ACIL: Analytic Class-Incremental Learning with Absolute Memorization and Privacy Protection. NIPS20.

---

### Author Response · Authors · 2024-11-21
**General Response**

Thank you to all the reviewers for your valuable feedback. We truly appreciate your time and effort. It seems there is a common misunderstanding regarding the definition of 'concept,' and we would like to clarify this in the following.

**Definition of Concept**

In our work, the concept is represented at the neuron level [1], where the neuron's activation value corresponds to the concept score, rather than the concept itself. To clarify, as discussed in Line 187, and following the work of [2, 3, 4] (specifically the Activation Maximization problem outlined in Sec. 3 of [2], Eq. 1 of [3], and Eq. 1 of [4]), we define a concept $c$ as the input that maximizes the activation value of the corresponding neuron in the neural network. Specifically, we denote our model as $g = h \cdot f$, where $h$ represents the classifier head and $f$ is the encoder. For a given concept $c_i$, we define it as $\text{argmax}_{c_i} f_i(c_i)$, where $f_i$  is the i-th element in $f(c_i)$. Hence, the number of concepts is equal to the dimensionality of the feature space.

The reason we define 'concept' at the neuron level is that previous studies [2, 3, 4, 5, 6, 7] have shown that the activation patterns of neurons in neural networks capture shared characteristics within input images. A notable example is shown in Fig. 6 of [5], which highlights the diverse visual concepts that neurons can discern. While the idea of a 'neuron concept' is not universally defined, and our approach is just one interpretation, there is broad consensus that neurons are indeed capable of representing such concepts. This property is essential for neural networks to perform perceptual tasks.

**Derivable and Underivable Concept**

If we train all the data in a supervised manner, then we can get high-quality known class concepts $C^k$ and novel class concepts $C^u$. However, due to the absence of label information, it is hard to acquire novel class concepts $C^u$. Nevertheless, the semantic relationship between novel and known classes in GCD problems suggests that some novel class concepts are linked to known class concepts. Therefore, we believe these novel class concepts can be generated from known class concepts: Given $C^k$ and $C^u, \exists$ function $g$ and a subset $C^g \subseteq C^u$, s.t $C^g = g(C^k)$. Therefore,  all class concepts $C = C^k \bigcup C^u$ can be divided into two groups, those derivable from known class concepts (derivable concepts) and those that are not (underivable concepts). In our paper, due to the relationship between derivable concepts and known class concepts, we believe the concept scores of derivable concepts can be generated based on the known class concept scores. Therefore, we utilize a linear layer as our generator layer, to generate derivable concept scores based on the known concept scores. Because our 'concepts' are directly linked to the corresponding neuron, by doing so, we can generate derivable concepts. The visualization of derivable and underivable concepts is also provided in Appendix M.

**Derivable Concepts’ Utility**

We believe that known classes can facilitate the learning of novel classes primarily due to the shared conceptual similarities between them. The experiments presented in Table 1 demonstrate that focusing on combining known class concepts can yield excellent results, thereby validating the significance of derivable concepts. Our approach aligns with the premises of references [8] and [9]: [8] posits that NCD is solvable when there is a high degree of semantic similarity between known and unknown classes, while [9] introduces a metric to quantify semantic similarity. Unlike these works, our study adopts a more experimental analytical perspective, employing a straightforward method to leverage high-level semantic information.

We sincerely hope the above response can address the concerns related to the concept. If not, please feel free to raise any remaining issues, and we will be more than happy to address them.


[1] Lee J H, et al. From neural activations to concepts: A survey on explaining concepts in neural networks.

[2] Erhan D, et al. Visualizing higher-layer features of a deep network.

[3] Nguyen A, et al. Understanding neural networks via feature visualization: A survey[J]. Explainable AI: interpreting, explaining and visualizing deep learning.

[4] Nguyen A, et al. Synthesizing the preferred inputs for neurons in neural networks via deep generator networks

[5] D. Bau et al, Network Dissection: Quantifying Interpretability of Deep Visual Representations

[6] Zeyuan Allen-Zhu et al. Towards understanding ensemble, knowledge distillation and self-distillation in deep learning.

[7] Matthew D Zeiler et al. Visualizing and understanding convolutional networks.

[8] Meta Discovery: Learning to Discover Novel Classes Given Very Limited Data

[9] Supervised Knowledge May Hurt Novel Class Discovery Performance

---

> ### Comment · Reviewer_wv58 · 2024-11-21
> **[Solved] Reminder by Reviewer wv58: The Authors deleted the concern I raised regarding a point in their initial Global Response without providing any clarification or notification**
>
> **I observed that, after I highlighted and disagreed the following quoted statement from the Authors' initial Global Response, the Authors EDITED their Global Response and DELETED this part WITHOUT providing any clarification or notification:**
>
> > "**Concept and Semantic Visual Cues:** We represent 'concepts' as semantic visual cues in Fig. 2 just for understanding."
> > — Authors' initial Global Response, as recorded in the Revisions history. Can be found by clicking the `Revisions` button.
>
> I would like to bring this to the attention of other Reviewers, ACs, and Viewers as a reminder that my concern regarding this statement remains valid and can still be found in the initial Global Response through the revision history.
>
> **It is concerning that the Authors chose to remove this part of their response without addressing it explicitly, while I understand that this issue may influence the evaluation of the paper. Such actions may mislead ACs and other Reviewers about the entire review process and the concerns raised.**
>
> This situation highlights the advantage of a fully open-review system: it ensures transparency and traceability. I strongly encourage the Authors to engage directly with reviewer feedback, addressing doubts and concerns constructively rather than avoiding or obscuring them.
>
> Reviewer wv58
>
> ******* After Author's clarification
>
> The Authors clarified that, this part of their response was deleted in the global response, but moved to my individual response, while I was writing my response based on their initial version. This is a misunderstanding caused by the not updated response when I was writing. **THE ABOVE CONCERN IS WELL-SOLVED NOW after Author's clarification.**  Thus, the above is not a concern anymore.

---

> > ### Author Response · Authors · 2024-11-21
> > **Clarify of the modification**
> >
> > Thank you for your timely feedback. However, I would like to clarify that the modification was made before your comment. Please carefully check the timestamp of each comment.
> >
> > The change was implemented because we recognized that the issue was specific to your concern, rather than a general problem.
> >
> > We have not removed the related response but have instead moved it to your part. Please let us know if you have any further concerns.

---

> > > ### Comment · Reviewer_wv58 · 2024-11-21
> > > **Thank you for the clarification!**
> > >
> > > Thank you for the clarification Authors! I wrote my response based on your initial invidual response and global response. Therefore, I did not notice you moved that part to my individual response, while I was writing.
> > >
> > > It's clear now. Thank you for the clarification!

---

### Meta-Review · Area_Chair_vUNa · 2024-12-17

**Metareview:**

This paper introduces a concept learning framework for generalized category discovery (GCD) problem, which is based on the analysis experiments. The paper initially got three negative scores.

The main strengths include: 1) novel concept-based architecture; 2) well-written; 3) strong performance; 4) comprehensive discussion on related works.

The authors have provided a rebuttal. After checking the rebuttal and comments of other reviewers, the reviewers agreed that most of the concerns have solved, but there are still several important drawback remained, include: 1) novelty is somewhat limited, as the frameworks have been widely used in the GCD tasks; 2) unclear definition: the definition of concept is not clearly provided, which is very important in this paper; and 3) unclear explanation of relation to feature space. Considering these remaining drawbacks, the AC thinks this paper cannot meet the requirement of ICLR at this point and thus regrets to recommend rejection.

**Additional Comments On Reviewer Discussion:**

The authors have provided a rebuttal. The reviewers have checked the rebuttal and comments from others. However, the reviewer still think the main drawbacks are not well-solved and thus keep their original negative ratings.

---

### Decision · Program_Chairs · 2025-01-22

Reject